

# Exploring a high-level programming model for the NWP domain using ECMWF microphysics schemes

Stefano Ubbiali[1], Christian Kühnlein[2], Christoph Schär[1], Linda Schlemmer[3], Thomas C. Schulthess[4,5], Michael Staneker[2], and Heini Wernli[1]

[1]Institute for Atmospheric and Climate Science (IAC), ETH Zürich, Switzerland
[2]European Centre for Medium-Range Weather Forecasts (ECMWF), Bonn, Germany
[3]Deutscher Wetterdienst (DWD), Offenbach, Germany
[4]Institute for Theoretical Physics (ITP), ETH Zürich, Switzerland
[5]Swiss National Supercomputing Centre (CSCS), Lugano, Switzerland

**Correspondence:** Stefano Ubbiali (subbiali@phys.ethz.ch)

**Abstract.** We explore the domain-specific Python library GT4Py (GridTools for Python) for implementing a representative physical parametrization scheme and the related tangent-linear & adjoint algorithms from the Integrated Forecasting System (IFS) of ECMWF. GT4Py encodes stencil operators in an abstract and hardware-agnostic fashion, thus enabling more concise, readable and maintainable scientific applications. The library achieves

high performance by translating the application into targeted low-level coding implementations. Here, the main goal is to study the correctness and performance-portability of the Python rewrites with GT4Py against the reference Fortran code and a number of automatically and manually ported variants created by ECMWF. The present work is part of a larger cross-institutional effort to port weather and climate models to Python with GT4Py. The focus of the current work is the IFS prognostic cloud microphysics scheme, a core physical parametrization represented by

a comprehensive code that takes a significant share of the total forecast model execution time. In order to verify GT4Py for Numerical Weather Prediction (NWP) systems, we put additional emphasis on the implementation and validation of the tangent-linear and adjoint model versions which are employed in data assimilation. We benchmark all prototype codes on three European supercomputers characterized by diverse GPU and CPU hardware, node designs, software stacks and compiler suites. Once the application is ported to Python with GT4Py, we find excellent

portability, competitive performance, and robust execution in all tested scenarios including with reduced precision.





# 1 Introduction

Soon after its first public release in 1957, Fortran has become the language of choice for weather and climate models (Méndez et al., 2014). On the one hand, its functional programming style and built-in support for multi-dimensional arrays has granted Fortran large popularity in the whole scientific computing community. On the other, its low-level

nature guarantees fast execution of intensive mathematical operations on vector machines and conventional Central Processing Units (CPUs). In the last decades, these characteristics have permitted to run weather forecasts several times per day under tight operational schedules on High-Performance Computing (HPC) systems (Neumann et al., 2019).

In recent years, in response to the simultaneous end of Moore's law and Dennard scaling, and due to the societal

challenge to reduce energy consumption, the computer hardware landscape has been undergoing a rapid specialization to prevent unsustainable growth of the power envelope (Müller et al., 2019). As a result, most supercomputers nowadays have a heterogeneous node design, where energy-efficient accelerators such as Graphics Processing Units (GPUs) co-exist with traditional CPUs. Because Fortran has been conceived with CPU-centric machines in mind, efficient programming of hybrid HPC platforms using the core Fortran language can be challenging (Méndez et al.,

2014; Lawrence et al., 2018). Indeed, the sustained performance of legacy weather and climate model codes written in Fortran has decreased over the decades (Schulthess et al., 2018), revealing the urgency for algorithmic and software adaptations to remain competitive in the medium and long term (Bauer et al., 2021).

Compiler directives (or pragmas) are an attractive solution for parallelization, both to spread a workload across multiple CPU threads, and to offload data and computations to GPU. The most famous incarnations of this

programming paradigm are OpenMP (Dagum and Menon, 1998) and OpenACC (Chandrasekaran and Juckeland, 2017). Because compiler directives accommodate incremental porting and enable non-disruptive software development workflows, they are adopted by many weather and climate modeling groups, who are facing the grand challenge of accelerating large code-bases with thousands of source files and millions of lines of code, which stem from decades of scientific discoveries and software developments (Lapillonne et al., 2017, 2020; Randall et al., 2022). In order not to

threaten the overall readability of the code by exposing low-level instructions, the annotation of Fortran codes with compiler directives can be automated in the pre-processor step of the compilation process using tools such as the CLAW compiler (Clement et al., 2019) or the ECMWF source-to-source translation tool Loki[1]. Although pragma-based programming models can support intrusive hardware-specific code transformations, additional specialized

---

[1]github.com/ecmwf-ifs/loki





optimizations may still be required, which could finally lead to code duplication and worsen maintainability (Dahm
et al., 2023). Moreover, performance and portability are much dependent on the level of support and optimization
offered by the compiler stack.

On the contrary, domain-specific languages (DSLs) separate the code describing the science from the code actually
executing on the target hardware, thus enabling *performance-portability*, namely application codes that achieve
near-optimal performance on a variety of computer architectures (Deakin et al., 2019). Large portions of many
modeling systems are being rewritten using multiple and diverse DSLs, not necessarily embedded in Fortran. For
instance, the dynamical core of the weather prediction model from the COnsortium for Small-scale MOdeling
(COSMO; Baldauf et al., 2011) has been rewritten in C++ using the GridTools library (Afanasyev et al., 2021) to
port stencil-based operators to GPUs (Fuhrer et al., 2014, 2018). Similarly, HOMMEXX-NH (Bertagna et al., 2020)
is an architecture-portable C++ implementation of the non-hydrostatic dynamical core of the Energy Exascale Earth
System model (E3SM; Taylor et al., 2020) harnessing the Kokkos library to express on-node parallelism (Edwards
et al., 2014). The GungHo project for a new dynamical core at the UK Met Office blends the LFRic infrastructure
with the PSyclone code generator (Adams et al., 2019). Pace (Ben-Nun et al., 2022; Dahm et al., 2023) is a Python
rewrite of the Finite-Volume Cubed-Sphere Dynamical Core (FV3; Harris and Lin, 2013) using GT4Py to accomplish
performance-portability and productivity. Similarly, various Swiss partners including MeteoSwiss, ETH Zurich and
CSCS are porting the ICOsahedral Non-hydrostatic modeling framework (ICON; Zängl et al., 2015) to GT4Py (Luz
et al., 2024). In another related project (Kühnlein et al., 2023), a next-generation model for the IFS at ECMWF is
developed in Python with GT4Py building on FVM (Smolarkiewicz et al., 2016; Kühnlein et al., 2019).

The focus of the portability efforts mentioned above is the model dynamical core - the part of the model solving
numerically the fundamental nonlinear fluid-dynamics equations. In the present work, we turn the attention to
physical parametrizations - which account for the representation of subgrid-scale processes - and additionally address
the associated tangent-linear and adjoint algorithms. Parametrizations are being commonly ported to accelerators
using OpenACC (e.g., Fuhrer et al., 2014; Yang et al., 2019; Kim et al., 2021). Wrappers around low-level legacy
physics codes might then be designed to facilitate adoption within higher-level workflows (Monteiro et al., 2018;
McGibbon et al., 2021). Lately, first attempts at refactoring physical parametrizations with respect to portability
have been documented in the literature. For instance, Watkins et al. (2023) presented a rewrite of the MPAS-Albany
Land Ice (MALI) ice-sheet model using Kokkos. Here, we present a Python implementation of the cloud microphysics





schemes CLOUDSC and CLOUDSC2, which are part of the physics suite of the IFS at ECMWF[2]. Details on the formulation and validation of the schemes are discussed in Section 2. The proposed Python implementations build upon the GT4Py toolchain, and in the remainder of the paper we use the term CLOUDSC-GT4Py to refer to the

GT4Py rewrite of CLOUDSC, while the GT4Py ports of the nonlinear, tangent-linear and adjoint formulations of CLOUDSC2 are collectively referred to as CLOUDSC2-GT4Py. The working principles of the GT4Py framework are illustrated in Section 3, where we also advocate the advantages offered by domain-specific software approaches. Section 4 sheds some light on the infrastructure code, and how it can enable composable and reusable model components. In Section 5, we compare the performance of CLOUDSC-GT4Py and CLOUDSC2-GT4Py, as measured on three

leadership-class GPU-equipped supercomputers, to established implementations in Fortran and C/C++. We conclude the paper with final remarks and future development paths.

## 2   Defining the targeted scientific applications

Several physical and chemical mechanisms occurring in the atmosphere are active on spatial scales that are significantly smaller than the highest affordable model resolution. It follows that these mechanisms cannot be properly captured

by the resolved model dynamics, but need to be *parametrized*. Parametrizations express the bulk effect of subgrid-scale phenomena on the resolved flow in terms of the grid-scale variables. The equations underneath physical parametrizations are based on theoretical and semi-empirical arguments, and their numerical treatment commonly adheres to the *single-column* abstraction, so that adjustments can only happen within individual columns, with no data dependencies between columns. The atmospheric module of the IFS includes parametrizations dealing with

the radiative heat transfer, deep and shallow convection, clouds and stratiform precipitation, surface exchange, turbulent mixing in the planetary boundary layer, subgrid-scale orographic drag, non-orographic gravity wave drag, and methane oxidation (ECMWF, 2023).

The focus of this paper is on the cloud microphysics modules of the ECMWF: the CLOUDSC – used in operational forecasting – and the CLOUDSC2 – employed in the data assimilation. The motivation is three-fold: (i) both schemes

are among the most computationally expensive parametrizations, with the CLOUDSC accounting for up to 10% of the total execution time of the high-resolution operational forecasts at ECMWF; (ii) they are representative of the computational patterns ubiquitous in physical parametrizations; and (iii) they already exist in the form of *dwarfs*. The weather and climate "computational dwarfs", or simply "dwarfs", are model components shaped into

---

[2]As we mention in Section 2, the versions of CLOUDSC and CLOUDSC2 considered in this study correspond to older release cycles of the IFS than the one currently used in production.





stand-alone software packages to serve as archetypes of relevant computational motifs (Müller et al., 2019) and

provide a convenient platform for performance optimizations and portability studies (Bauer et al., 2020). In recent

years, the Performance and Portability Team of ECMWF created the CLOUDSC and CLOUDSC2 dwarfs. The

original Fortran codes for both packages, corresponding respectively to the IFS Cycle 41r2 and 46r1, have been

pulled out of the IFS codebase, slightly polished and finally made available in public code repositories[3]. Later, the

repositories have been enriched with alternative coding implementations, using different languages and programming

paradigms; the most relevant implementations will be discussed in Section 5.

### 2.1  CLOUDSC: Cloud microphysics of the forecast model

The CLOUDSC is a single-moment cloud microphysics scheme that parametrizes stratiform clouds and their

contribution to surface precipitation (ECMWF, 2023). It was implemented in the IFS Cycle 36r4 and has been

operational at ECMWF since November 2010. Compared to the pre-existing scheme, it accounts for five prognostic

variables (cloud fraction, cloud liquid water, cloud ice, rain and snow) and brings substantial enhancements in

different aspects, including treatment of mixed-phase clouds, advection of precipitating hydrometeors (rain and snow),

physical realism, and numerical stability (Nogherotto et al., 2016). For a comprehensive description of the scheme,

we refer the reader to Forbes et al. (2011) and the references therein. For all the coding versions considered in this

paper, including the novel Python rewrite, the calculations are validated by direct comparison of the output against

serialized language-agnostic reference data provided by ECMWF.

### 2.2  CLOUDSC2: Cloud microphysics in the context of data assimilation

The CLOUDSC2 scheme represents a streamlined version of CLOUDSC, devised for use in the four-dimensional

variational assimilation (4D-Var) at ECMWF (Courtier et al., 1994). 4D-Var merges short-term model integrations

with observations over a twelve-hour assimilation window to determine the best possible representation of the current

state of the atmosphere. This then provides the initial conditions for longer-term forecasts (Janisková and Lopez,

2023). The optimal synthesis between model and observational data is found by minimizing a cost function, which

is evaluated using the *tangent-linear* of the *non-linear* forecasting model, while the *adjoint* model is employed to

compute the gradient of the cost function (Errico, 1997; Janisková et al., 1999). For the sake of computational

economy, the tangent-linear and adjoint operators are derived from a simplified and regularized version of the full

non-linear model. The CLOUDSC2 is one of the physical parametrizations included in the ECMWF's simplified model,

---

[3]https://github.com/ecmwf-ifs/dwarf-p-cloudsc and https://github.com/ecmwf-ifs/dwarf-p-cloudsc2-tl-ad





together with radiation, vertical diffusion, orographic wave drag, moist convection, and non-orographic gravity wave activity (Janisková and Lopez, 2023). In the following, we provide a mathematical and algorithmic representation of the tangent-linear and adjoint versions of CLOUDSC2. For the sake of brevity, in the rest of the paper we will refer to the non-linear, tangent-linear and adjoint formulations of CLOUDSC2 using CLOUDSC2NL, CLOUDSC2TL and

CLOUDSC2AD, respectively.

Let $F : \boldsymbol{x} \mapsto \boldsymbol{y}$ be the functional description of CLOUDSC2, connecting the input fields $\boldsymbol{x}$ with the output variables $\boldsymbol{y}$. The tangent-linear operator $F'$ of $F$ is derived from the Taylor series expansion

$$F\left(\boldsymbol{x} + \delta\boldsymbol{x}\right) = \boldsymbol{y} + \delta\boldsymbol{y} = F\left(\boldsymbol{x}\right) + F'\left[\boldsymbol{x}\right]\left(\delta\boldsymbol{x}\right) + \mathcal{O}\left(||\delta\boldsymbol{x}||^2\right), \tag{1}$$

where $\delta\boldsymbol{x}$ and $\delta\boldsymbol{y}$ are variations on $\boldsymbol{x}$ and $\boldsymbol{y}$, and $||\cdot||$ is a suitable norm. The formal correctness of the coding

implementation of $F'$ can be assessed through the Taylor test (also called the "V-shape" test), which ensures that the following condition is satisfied up to machine precision:

$$\lim_{\lambda \to 0} \frac{F\left(x + \lambda\delta\boldsymbol{x}\right) - F\left(\boldsymbol{x}\right)}{F'\left[\boldsymbol{x}\right]\left(\lambda\delta\boldsymbol{x}\right)} = 1 \qquad \forall \boldsymbol{x}, \delta\boldsymbol{x}. \tag{2}$$

The logical steps carried out in the actual implementation of the Taylor test are sketched in Algorithm 1.

The adjoint operator $F^*$ of $F'$ is defined such that for the inner product $< \cdot, \cdot >$:

$$< \delta\boldsymbol{x}, F^*\left[\boldsymbol{y}\right]\left(\delta\boldsymbol{y}\right) > \; = \; < \delta\boldsymbol{y}, F'\left[\boldsymbol{x}\right]\left(\delta\boldsymbol{x}\right) > \qquad \forall \boldsymbol{x}, \delta\boldsymbol{x}, \boldsymbol{y}, \delta\boldsymbol{y}. \tag{3}$$

In particular, (3) must hold for $\boldsymbol{y} = F\left(\boldsymbol{x}\right)$ and $\delta\boldsymbol{y} = F'\left[\boldsymbol{x}\right]\left(\delta\boldsymbol{x}\right)$:

$$< \boldsymbol{x}, F^*\left[F\left(\boldsymbol{x}\right)\right]\left(F'\left[\boldsymbol{x}\right]\left(\delta\boldsymbol{x}\right)\right) > \; = \; < F'\left[\boldsymbol{x}\right]\left(\delta\boldsymbol{x}\right), F'\left[\boldsymbol{x}\right]\left(\delta\boldsymbol{x}\right) > \qquad \forall \boldsymbol{x}, \delta\boldsymbol{x}. \tag{4}$$

The latter condition is at the hearth of the so-called symmetry test for $F^*$ (see Algorithm 2).





---

**Algorithm 1** The Taylor test assessing the formal correctness of the coding implementation of the tangent-linear formulation of CLOUDSC2, denoted as CLOUDSC2TL. The three-dimensional arrays **x** and **y** collect the grid point values for all *nin* input fields and *nout* output fields of CLOUDSC2, respectively. The corresponding variations are $\delta\mathbf{x}$ and $\delta\mathbf{y}$. The grid consists of *ncol* columns, each containing *nlev* vertical levels. Note that compared to its functional counterpart $F'[\boldsymbol{x}] : \delta\boldsymbol{x} \mapsto \delta\boldsymbol{y}$, CLOUDSC2TL($\mathbf{x}, \delta\mathbf{x}$) returns both **y** and $\delta\mathbf{y}$. The coding implementation of the non-linear CLOUDSC2 is indicated as CLOUDSC2NL.

---

1: **function** TOTALNORM($ncol, nlev, nout, \mathbf{y}, \mathbf{y}_j, \delta\mathbf{y}_j$)          ▷ $\mathbf{y}, \mathbf{y}_j, \delta\mathbf{y}_j \in \mathbb{R}^{ncol \times nlev \times nout}$

2:     $total\_norm \leftarrow 0$

3:     $total\_count \leftarrow 0$

4:     **for** $l \leftarrow 1$ **to** $nout$ **do**

5:        $\beta \leftarrow \left| \sum_{i=1}^{nlev} \sum_{k=1}^{ncol} \delta\mathbf{y}_j(i, k, l) \right|$

6:        **if** $\beta > 0$ **then**

7:           $total\_norm \leftarrow total\_norm + \left| \sum_{i=1}^{nlev} \sum_{k=1}^{ncol} (\mathbf{y}_j(i, k, l) - \mathbf{y}(i, k, l)) \right| / \beta$

8:           $total\_count \leftarrow total\_count + 1$

9:     **if** $total\_count > 0$ **then**

10:       **return** $total\_norm \,/\, total\_count$

11:     **else**

12:       **return** $0$

13: **procedure** TAYLORTEST($ncol, nlev, nin, nout, \mathbf{x}$)          ▷ $\mathbf{x} \in \mathbb{R}^{ncol \times nlev \times nin}$

14:     $\delta\mathbf{x} \leftarrow 0.01 * \mathbf{x}$

15:     $(\mathbf{y}, \delta\mathbf{y}) \leftarrow$ CLOUDSC2TL($\mathbf{x}, \delta\mathbf{x}$)          ▷ $\mathbf{y}, \delta\mathbf{y} \in \mathbb{R}^{ncol \times nlev \times nout}$

16:     $norms \leftarrow ()$

17:     $jstart \leftarrow 1$

18:     **for** $j \leftarrow 1$ **to** $10$ **do**

19:        $\mathbf{y}_j \leftarrow$ CLOUDSC2NL($\mathbf{x} + 10^{-j} * \delta\mathbf{x}$)

20:        $norms \leftarrow norms \cup \left(1 - \text{TOTALNORM}(ncol, nlev, nout, \mathbf{y}, \mathbf{y}_j, 10^{-j} * \delta\mathbf{y})\right)$

21:        **if** $jstart = 1$ & $norms(j) < 0.5$ **then**

22:           $jstart \leftarrow j$

23:     $test \leftarrow -10$

24:     $negat \leftarrow$ True

25:     **for** $j \leftarrow jstart$ **to** $9$ **do**

26:        **if** $negat$ & $norms(j + 1) \geq norms(j)$ **then**

27:           $test \leftarrow test + 10$

28:        $negat \leftarrow norms(j + 1) < norms(j)$

29:     **if** $test = -10$ **then**

30:       $test \leftarrow 11$

31:     **if** $\min_{jstart \leq j \leq 10}(norms(j)) > 10^{-5}$ **then**

32:       $test \leftarrow test + 7$

33:     **if** $\min_{jstart \leq j \leq 10}(norms(j)) > 10^{-6}$ **then**

34:       $test \leftarrow test + 5$

35:     **if** $test \leq 5$ **then**

36:       **print** *"The Taylor test passed."*

37:     **else**

38:       **print** *"The Taylor test failed."*

---





---

**Algorithm 2** The symmetry test assessing the formal correctness of the coding implementation of the adjoint formulation of CLOUDSC2, denoted as CLOUDSC2AD. The machine epsilon is indicated as $\varepsilon$; all other symbols have the same meaning as in Algorithm 1. Note that compared to its functional counterpart $F^*[F(\boldsymbol{x})]: \delta\boldsymbol{y} \mapsto \delta\boldsymbol{x}^*$, CLOUDSC2AD$(\mathbf{x}, \delta\mathbf{y})$ returns both $\mathbf{y}$ and $\delta\mathbf{x}^*$.

---

1: **function** COLUMNWISEINNERPRODUCT($ncol, nlev, ndim, \mathbf{a}, \mathbf{b}$)             $\triangleright\ \mathbf{a}, \mathbf{b} \in \mathbb{R}^{ncol \times nlev \times ndim}$

2:      $\boldsymbol{c} \leftarrow \mathbf{0} \in \mathbb{R}^{ncol}$

3:      **for** $l \leftarrow 1$ **to** $ndim$ **do**

4:          **for** $i \leftarrow 1$ **to** $ncol$ **do**

5:              $\boldsymbol{c}(i) \leftarrow \boldsymbol{c}(i) + \sum_{k=1}^{ncol} \mathbf{a}(i,k,l) * \mathbf{b}(i,k,l)$

6:      **return** $\boldsymbol{c}$

 

7: **procedure** SYMMETRYTEST($ncol, nlev, nin, nout, \mathbf{x}, \varepsilon$)             $\triangleright\ \mathbf{x} \in \mathbb{R}^{ncol \times nlev \times nin}$

8:      $\delta\mathbf{x} \leftarrow 0.01 * \mathbf{x}$

9:      $(\mathbf{y}, \delta\mathbf{y}) \leftarrow$ CLOUDSC2TL$(\mathbf{x}, \delta\mathbf{x})$             $\triangleright\ \mathbf{y}, \delta\mathbf{y} \in \mathbb{R}^{ncol \times nlev \times nout}$

10:      $\boldsymbol{c_y} \leftarrow$ COLUMNWISEINNERPRODUCT($ncol, nlev, nout, \delta\mathbf{y}, \delta\mathbf{y}$)

11:      $(\mathbf{y}, \delta\mathbf{x}^*) \leftarrow$ CLOUDSC2AD$(\mathbf{x}, \delta\mathbf{y})$             $\triangleright\ \mathbf{x}^*, \delta\mathbf{x}^* \in \mathbb{R}^{ncol \times nlev \times nin}$

12:      $\boldsymbol{c_x} \leftarrow$ COLUMNWISEINNERPRODUCT($ncol, nlev, nin, \delta\mathbf{x}, \delta\mathbf{x}^*$)

13:      $success \leftarrow$ True

14:      **for** $i \leftarrow 1$ **to** $ncol$ **do**

15:          **if** $\boldsymbol{c_x}(i) = 0$ **then**

16:              $c \leftarrow |\boldsymbol{c_y}(i)| / \varepsilon$

17:          **else**

18:              $c \leftarrow |\boldsymbol{c_y}(i) - \boldsymbol{c_x}(i)| / |\varepsilon * \boldsymbol{c_x}(i)|$

19:          $success \leftarrow success \ \&\ c < 10^3$

20:      **if** $success$ **then**

21:          **print** *"The symmetry test passed."*

22:      **else**

23:          **print** *"The symmetry test failed."*

---

## 3   A domain-specific approach to scientific software development

In scientific software development, it is common practice to conceive a first proof-of-concepts implementation of a numerical algorithm in a high-level programming environment like MATLAB/Octave (Lindfield and Penny, 2018), or Python. Because these languages do not require compilation and support dynamic typing, they provide a breeding ground for fast prototyping. However, the direct adoption of interpreted languages in HPC has historically been hindered by their intrinsic slowness. To squeeze more performance out of the underlying silicon, the initial proof-

of-concept is translated into either Fortran, C or C++. This leads to the so-called "two-language problem", where the programming language used for the germinal prototyping is abandoned in favor of a faster language that might be more complicated to use. The lower-level code can be parallelized for shared memory platforms using OpenMP directives, while distributed memory machines can be targeted using Message Passing Interface (MPI) libraries. The

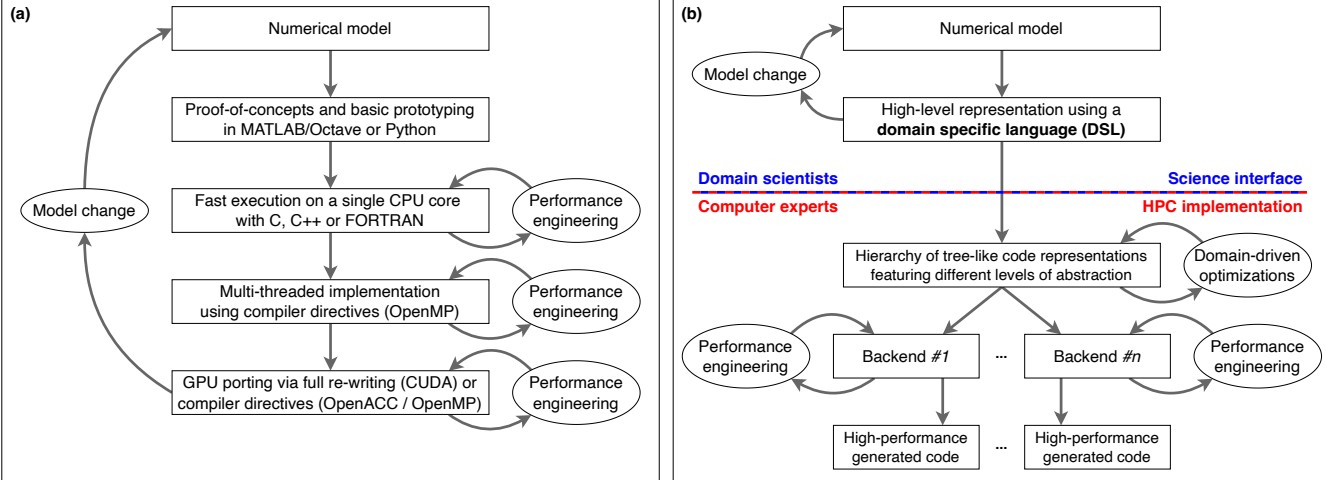

**Figure 1.** Diagrams comparing **(a)** a well-established workflow in scientific software development, and **(b)** a DSL-based approach resembling the software engineering strategy advocated in this paper. The red-and-blue dashed line in **(b)** mark the separation-of-concerns between the domain scientists and the computer experts.

resulting code can later be migrated to GPUs, offering outstanding compute throughput and memory bandwidth

especially for Single Instruction Multiple Data (SIMD) applications. GPU porting is accomplished using either OpenACC or OpenMP directives, or via a CUDA[4] or HIP[5] rewriting, amongst others. To efficiently run the model at scale on multiple GPUs, a GPU-aware MPI build should be chosen, so to possibly avoid costly memory transfers between host and device and better overlap computations and communications.

The schematic visualization in Fig. 1a highlights how the above workflow leads to multiple implementations of the

same scientific application utilizing different programming models and coding styles. This unavoidably complicates software maintainability: ideally, any modification in the numerical model should be encoded in all implementations, so to preserve the coherency across the hierarchy. The maintainability problem is exacerbated as the number of lines of code, the pool of platforms to support, and the user-base increase. This situation has been known as the "software productivity gap" (Lawrence et al., 2018), and we argue that it cannot be alleviated by relying on general-purpose

programming paradigms and monolithic code designs. Instead, it calls for a more synergistic collaboration between domain scientists (which here include model developers, weather forecasters, and weather and climate scientists) and computer experts. A path forward is provided by DSLs through *separation of concerns* (Fig. 1b), so that domain scientists can express the science using syntactic constructs that are aligned with the semantics of the application domain and hide any architecture-specific detail. The resulting source code is thus hardware-agnostic, more concise,

---

[4]https://docs.nvidia.com/cuda/
[5]https://rocm.docs.amd.com/projects/HIP/en/latest/



easier to read, and easier to manipulate. A toolchain developed by software engineers then employs automatic code

generation techniques to synthesize optimized parallel code for the target computer architecture in a transparent fashion.

## 3.1 The GT4Py framework

GT4Py[6] is a Python library to generate high-performance implementations of stencil[7] kernels as found in weather

and climate applications. The library is developed and maintained by the Swiss National Supercomputing Center (CSCS), ETH Zurich, and the Swiss Federal Office of Meteorology and Climatology (MeteoSwiss), and benefits from important contributions by international partners such as the Paul Allen Institute for Artificial Intelligence ($AI^2$). The choice of embedding the GT4Py framework in Python has been mainly dictated by the following factors: (i) Python is taught in many academic courses due its clean, intuitive and expressive syntax, so that a significant fraction of

early-career domain scientists is exposed to the language; (ii) it admits a powerful ecosystem of open source packages for building end-to-end applications; (iii) it is possible to seamlessly interface Python with lower-level languages with minimal overhead and virtually no memory copies; (iv) under the thrust of the Artificial Intelligence and Machine Learning community (AI/ML), the popularity and adoption of Python across the whole scientific community is constantly growing, as opposed to Fortran (Shipman and Randles, 2023). The proposed Python implementations of

CLOUDSC and CLOUDSC2 are based on the first public release of GT4Py, which only supports Cartesian grids. Latest advancements to support unstructured meshes (contained in the sub-package `gt4py.next`) are not discussed in this study.

Figure 2 showcases the main steps undertaken by the GT4Py toolchain to translate the high-level definition of the horizontal Laplacian operator into optimized code, which can be directly called from within Python. The

stencil definition is given as a regular Python function using the GTScript DSL. GTScript abstracts for-loops away: computations are described for a single point of a three-dimensional Cartesian grid, and can be differentiated for the vertical boundaries using the `interval` context manager. Vertical loops are replaced by `computation` contexts, which define the iteration order along the vertical axis: either `PARALLEL` (meaning no vertical data dependencies between horizontal planes), `FORWARD` or `BACKWARD`. Each assignment statement within a computation block can be

thought of as a loop over a horizontal plane; no horizontal data dependencies are allowed. Neighbouring points are

---

[6]https://github.com/GridTools/gt4py
[7]A *stencil* is an operator that computes array elements by accessing a fixed pattern of neighbouring items.





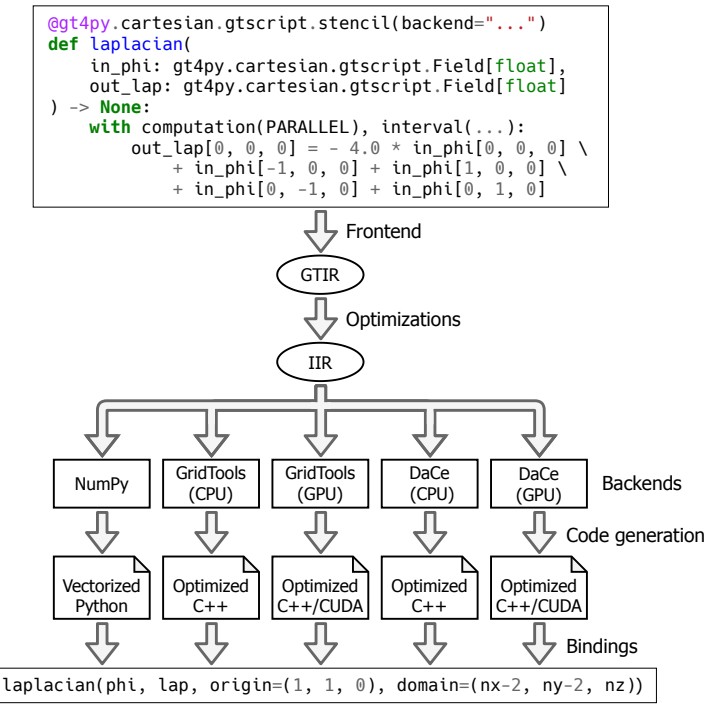

**Figure 2.** Simplified view on the internal stages carried out by the GT4Py toolchain to generate a high-performance CPU or GPU implementation of the horizontal Laplacian stencil starting from its GTScript definition. For the sake of visualization, only two intermediate representations (IRs) are included: the GridTools IR (GTIR) and the Implementation IR (IIR).

accessed through relative offsets, with the first two offsets being the horizontal offsets, and the last offset being the vertical offset.

Any function marked with the `gt4py.cartesian.gtscript.stencil` decorator is translated by the GT4Py *frontend* into a hierarchy of tree-like Intermediate Representations (IRs), featuring different levels of abstractions

to accommodate diverse optimizations and transformations (Gysi et al., 2021). The lowest-level IR (denoted as Implementation IR, or IIR) is consumed by the *backends* to generate code that is either optimized for a given architecture or suited to a specific purpose. The following backends are currently available:

– NumPy (Harris et al., 2020) is the *de facto* standard for array computing in Python, and can be used for debugging and fast-prototyping;

– GridTools (Afanasyev et al., 2021) is a set of libraries and utilities to write performance-portable applications in the area of weather and climate;



- DaCe (Ben-Nun et al., 2019) is a parallel programming framework, which internally uses the Stateful DataFlow multiGraph (SDFG) data-centric intermediate representation to decouple domain science and performance engineering.

The generated code is compiled under the hood, and Python bindings for the resulting executable are automatically produced, so that the stencil can finally be executed by passing the input and output fields and by specifying the origin and size of the computation domain. GT4Py provides convenient utilities to allocate arrays with an optimal memory layout for any given backend, relying on NumPy for CPU storages and CuPy (Nishino and Loomis, 2017) for GPU storages. Concerning GPU computing, we highlight that GT4Py supports both NVIDIA and AMD GPUs.

A more realistic and pertinent code sample is provided in Listing 1. It is an abridged GT4Py implementation of the procedure computing the saturation water vapor pressure as a function of air pressure and temperature. The code is extracted from the CLOUDSC2-GT4Py dwarf and highlights two additional features of GTScript: functions and external symbols. Functions can be thought of as macros, and can be used to improve composability, reusability and readability. External symbols are used to encode those scalar parameters (e.g. physical constants) that are kept

constant throughout a simulation, and might only change between different model setups. External values must be provided at stencil compilation time. The functionalities provided by the package `ifs_physics_common` will be discussed in the following section.

## 4   Infrastructure code

All stencils of CLOUDSC-GT4Py and CLOUDSC2-GT4Py are defined, compiled and invoked within classes that

leverage the functionalities provided by the Sympl package (Monteiro et al., 2018). Sympl is a toolset of Python utilities to write self-contained and self-documented model components. Because the components share a common Application Public Interface (API), they favor modularity, composability and inter-operability (Schär et al., 2019). These aspects are of utter importance, for instance, when it comes to assessing the impact of process coupling on weather forecasts and climate projections (Ubbiali et al., 2021).

Sympl components interact through dictionaries whose keys are the names of the model variables (fields), and whose values are xarray's `DataArray`s (Hoyer and Hamman, 2017) collecting the grid point values, the labelled dimensions, the axis coordinates, and the units for those variables. The most relevant component exposed by Sympl is `TendencyComponent`, producing tendencies for prognostic variables and retrieving diagnostics. The class defines a





---

**Listing 1** GTScript (the Python-embedded DSL exposed by GT4Py) functions and stencil computing the saturation water vapor pressure given the air pressure and temperature. Abridged excerpt from the CLOUDSC2-GT4Py dwarf.

```python
@gt4py.cartesian.gtscript.function
def foealfcu(t):
    from __externals__ import RTICECU, RTWAT, RTWAT_RTICECU_R
    return min(1.0, ((max(RTICECU, min(RTWAT, t)) - RTICECU) * RTWAT_RTICECU_R) ** 2)

@gt4py.cartesian.gtscript.function
def foeewmcu(t):
    from __externals__ import R2ES, R3IES, R3LES, R4IES, R4LES, RTT
    return R2ES * (
        foealfcu(t) * exp(R3LES * (t - RTT) / (t - R4LES))
        + (1 - foealfcu(t)) * (exp(R3IES * (t - RTT) / (t - R4IES)))
    )

@ifs_physics_common.framework.stencil.stencil_collection("saturation")
def saturation(
    in_ap: gtscript.Field[float], in_t: gtscript.Field[float], out_qsat: gtscript.Field[float]
):
    from __externals__ import LPHYLIN, QMAX, R2ES, R3IES, R3LES, R4IES, R4LES, RETV, RTT
    with computation(PARALLEL), interval(...):
        if LPHYLIN:  # linearized physics
            alfa = foealfa(in_t)
            foeewl = R2ES * exp(R3LES * (in_t - RTT) / (in_t - R4LES))
            foeewi = R2ES * exp(R3IES * (in_t - RTT) / (in_t - R4IES))
            foeew = alfa * foeewl + (1 - alfa) * foeewi
            qs = min(foeew / in_ap, QMAX)
        else:
            ew = foeewmcu(in_t)
            qs = min(ew / in_ap, QMAX)
        out_qsat[0, 0, 0] = qs / (1.0 - RETV * qs)
```

---

minimal interface to declare the list of input and output fields, and initialize and run an instance of the class. This
imposes minor constraints on model developers when writing a new physics package.

The bespoke infrastructure code for CLOUDSC-GT4Py and CLOUDSC2-GT4Py is bundled as an installable Python package called `ifs_physics_common`. Not only it builds upon Sympl, but is also extends it with grid-aware and stencil-oriented functionalities. Both the CLOUDSC cloud microphysics and the non-linear, tangent-linear and adjoint formulations of CLOUDSC2 are encoded as stand-alone `TendencyComponent` classes settled over a
`ComputationalGrid`. The latter is a collection of index spaces for different grid locations. For instance, `(I, J, K)` corresponds to cell centers, while `(I, J, K-1/2)` denotes vertically-staggered grid points. For any input and output field, its name, units and grid location are specified as class properties. When running the component via the



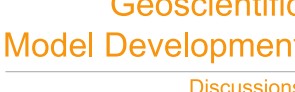

**Listing 2** A Python class to compute the saturation water vapor pressure given the air pressure and temperature. Abridged excerpt from the CLOUDSC2-GT4Py dwarf.

```python
import cupy as cp
from functools import cached_property
import numpy as np
from typing import Optional, Union
from ifs_physics_common.framework.components import DiagnosticComponent
from ifs_physics_common.framework.config import GT4PyConfig
from ifs_physics_common.framework.grid import ComputationalGrid, I, J, K

# type alias originally defined in ifs_physics_common.utils.typingx
StorageDict = dict[str, Union[cp.ndarray. np.ndarray]]

class Saturation(DiagnosticComponent):
    def __init__(
        self,
        computational_grid: ComputationalGrid,
        lphylin: bool,
        yoethf_parameters: Optional[dict[str, float]] = None,
        yomcst_parameters: Optional[dict[str, float]] = None,
        gt4py_config: GT4PyConfig,
    ) -> None:
        super().__init__(computational_grid, gt4py_config)
        externals = {"LPHYLIN": lphylin, "QMAX": 0.5}
        externals.update(yoethf_parameters or {})
        externals.update(yomcst_parameters or {})
        self.saturation = self.compile_stencil("saturation", externals)

    @cached_property
    def _input_properties(self):
        return {"ap": {"grid": (I, J, K), "units": "Pa"}, "t": {"grid": (I, J, K), "units": "K"}}

    @cached_property
    def _diagnostic_properties(self):
        return {"qsat": {"grid": (I, J, K), "units": "g g^-1"}}

    def array_call(self, state: StorageDict, out: StorageDict) -> None:
        self.saturation(
            in_ap=state["ap"],
            in_t=state["t"],
            out_qsat=out["qsat"],
            origin=(0, 0, 0),
            domain=self.computational_grid.grids[I, J, K].shape,
        )
```





*dunder* method `__call__`, Sympl transparently extracts the raw data from the input `DataArray`s according to the information provided in the class definition. This step may involve units conversion and axis transposition. The

resulting storages are forwarded to the method `array_call`, which carries out the actual computations, possibly by executing GT4Py stencil kernels.

Listing 2 brings a concrete example from CLOUDSC2-GT4Py: a model component leveraging the stencil defined in Listing 1 to compute the saturation water vapor pressure. The class inherits `DiagnosticComponent`, a stripped-down version of `TendencyComponent`, which only retrieves diagnostic quantities. Within the instance initializer `__init__`, the

stencil from Listing 1, registered using the decorator `ifs_physics_common.framework.stencil.stencil_collection`, is compiled using the utility method `compile_stencil`. The options configuring the stencil compilation (e.g. the GT4Py backend) are fetched from the dataclass `GT4PyConfig`.

## 5 Performance analysis

In this section, we highlight the results from a comprehensive performance testing. We compare the developed
CLOUDSC-GT4Py and CLOUDSC2-GT4Py codes against reference Fortran versions and various other programming prototypes. The simulations were performed on three different supercomputers:

(i) Piz Daint[8], an HPE Cray XC40/XC50 system installed at CSCS in Lugano, Switzerland;

(ii) MeluXina[9], an ATOS BullSequana XH2000 machine hosted by LuxConnect in Bissen, Luxembourg, and procured by the EuroHPC Joint Undertaking (JU) initiative;

(iii) the Cray HPE EX235a supercomputer LUMI[10], an EuroHPC pre-exascale machine at the Science Information Technology Center (CSC) in Kajaani, Finland.

On each machine, the CLOUDSC and CLOUDSC2 applications are executed on a single hybrid node, that sports one or multiple GPU accelerators alongside the host CPU. An overview of the node architectures for the three considered supercomputers can be found in Table 1.

Besides the GT4Py codes, we involve up to four alternative lower-level programming implementations, which will be documented in an upcoming publication.

(a) The baseline Fortran version, enriched with OpenMP directives for multi-threading execution on CPU.

---

[8] https://www.cscs.ch/computers/piz-daint
[9] https://docs.lxp.lu/
[10] https://docs.lumi-supercomputer.eu/





| System | CPU | GPU | RAM | NUMA domains |
|---|---|---|---|---|
| Piz Daint | 1x Intel Xeon E5-2690v3 12c | 1x NVIDIA Tesla P100 16GB | 64 GB | 1 |
| MeluXina | 2x AMD EPYC Rome 7452 32c | 4x NVIDIA Tesla A100 40GB | 512 GB | 4 |
| LUMI | 1x AMD EPYC Trento 7A53 64c | 4x AMD Instinct MI250X | 512 GB | 8 |

**Table 1.** Overview of the node architecture for the hybrid partition of Piz Daint, MeluXina and LUMI. Only the technical specifications which are most relevant for the purposes of this paper are reported.

(b) An optimized GPU-enabled version based on OpenACC using the single-column coalesced (SCC) loop layout in combination with loop fusion and temporary local array demotion (so-called "k-caching"). While the SCC loop layout yields more efficient access to device memory and increased parallelism, the k-caching technique significantly reduces register pressure and memory traffic. This is achieved via loop fusion to eliminate most loop-carried dependencies and consequently allows to demote temporaries to scalars.

(c) The currently best-performing Loki generated and GPU-enabled variant.

(d) An optimized GPU-enabled version of CLOUDSC including k-caching. The code is written either in CUDA or HIP, to target both NVIDIA GPUs (shipped with Piz Daint and MeluXina) and AMD GPUs (available on LUMI).

Table 2 documents the compiler specifications employed for each of the programming implementations, on Piz Daint, MeluXina and LUMI. We consistently apply the most aggressive optimization, ensuring that the underlying code manipulations do not harm validation. For the different algorithms at consideration, validation is carried out as follows.

– For CLOUDSC and CLOUDSC2NL, the results from each coding version are directly compared with serialized reference data produced on the CPU. For each output field, we perform an element-wise comparison using the NumPy function `allclose`[11]. Specifically, the GT4Py rewrites validate on both CPU and GPU with an absolute and relative tolerance of $10^{-12}$ and $10^{-18}$ when employing double precision. When reducing the precision to 32-bits, the absolute and relative tolerance levels need to be increased to $10^{-4}$ and $10^{-7}$ on CPU, and $10^{-2}$ and $10^{-7}$ on GPU. In the latter case, we observe that the field representing the enthalpy flux of ice still does not pass validation. We attribute the larger deviation from the baseline data on the device to the different instruction sets underneath CPUs and GPUs.

---

[11]https://numpy.org/devdocs/reference/generated/numpy.allclose.html





| | Implementation | CLOUDSC | CLOUDSC2: Non-linear | CLOUDSC2: Symmetry test |
|---|---|---|---|---|
| **Piz Daint** | Fortran: OpenMP (CPU) | Intel Fortran 2021.3.0 | Intel Fortran 2021.3.0 | Intel Fortran 2021.3.0 |
| | Fortran: OpenACC (GPU) | NVIDIA Fortran 21.3-0 | - | - |
| | Fortran: Loki (GPU) | NVIDIA Fortran 21.3-0 | NVIDIA Fortran 21.3-0 | - |
| | C: CUDA (GPU) | NVIDIA CUDA 11.2.67 | - | - |
| | GT4Py: CPU k-first | g++ (GCC) 10.3.0 | g++ (GCC) 10.3.0 | g++ (GCC) 10.3.0 |
| | GT4Py: DaCe (GPU) | NVIDIA CUDA 11.2.67 | NVIDIA CUDA 11.2.67 | NVIDIA CUDA 11.2.67 |
| **MeluXina** | Fortran: OpenMP (CPU) | NVIDIA Fortran 22.7-0 | NVIDIA Fortran 22.7-0 | - |
| | Fortran: OpenACC (GPU) | NVIDIA Fortran 22.7-0 | - | - |
| | Fortran: Loki (GPU) | NVIDIA Fortran 22.7-0 | NVIDIA Fortran 22.7-0 | - |
| | C: CUDA (GPU) | NVIDIA CUDA 11.7.64 | - | - |
| | GT4Py: CPU k-first | g++ (GCC) 11.3.0 | g++ (GCC) 11.3.0 | g++ (GCC) 11.3.0 |
| | GT4Py: DaCe (GPU) | NVIDIA CUDA 11.7.64 | NVIDIA CUDA 11.7.64 | NVIDIA CUDA 11.7.64 |
| **LUMI** | Fortran: OpenMP (CPU) | Cray Fortran 14.0.2 | Cray Fortran 14.0.2 | Cray Fortran 14.0.2 |
| | Fortran: OpenACC (GPU) | Cray Fortran 14.0.2 | - | - |
| | Fortran: Loki (GPU) | Cray Fortran 14.0.2 | Cray Fortran 14.0.2 | - |
| | C: HIP (GPU) | Cray C/C++ 15.0.1 | - | - |
| | GT4Py: CPU k-first | Cray C/C++ 15.0.1 | Cray C/C++ 15.0.1 | Cray C/C++ 15.0.1 |
| | GT4Py: DaCe (GPU) | Cray C/C++ 15.0.1 | Cray C/C++ 15.0.1 | Cray C/C++ 15.0.1 |

**Table 2.** For each coding version of the CLOUDSC and CLOUDSC2 dwarfs considered in the performance analysis, the table reports the compiler suite used to compile the codes on Piz Daint, MeluXina and LUMI. The codes are compiled with all major optimization options enabled. Those implementations which are either not available or not working are marked with a dash; more details, as well as a high-level description of each coding implementation, are provided in the text.

- All implementations of CLOUDSC2TL and CLOUDSC2AD are validated using the Taylor test (cf. Algorithm 1) and the symmetry test (cf. Algorithm 2), respectively. However, the conditions of both tests are not satisfied when using single precision. This is not surprising, since both tests are highly sensitive to round-off errors. Nevertheless, performance numbers for the execution of the algorithms were taken.

The source repositories for CLOUDSC and CLOUDSC2 dwarfs may include multiple variants of each reference implementation, varying for the optimization strategies. In our analysis, we always take into account the fastest variant of each alternative implementation; for the sake of reproducibility, Table 3 contains the strings identifying the coding versions at consideration and the corresponding NPROMA employed in the runs.

For the interpretation of the CPU versus GPU performance numbers, we note that host codes are executed on all the cores available on a single Non-Uniform Memory Access (NUMA) domain of a compute node, while device codes are launched on the GPU attached to that NUMA domain. In a distributed-memory context, this choice allows to fit the same number of MPI ranks per node, either on CPU or GPU. Table 1 reports the number of NUMA partitions per





|  | Implementation | CLOUDSC | CLOUDSC2: Non-linear | CLOUDSC2: Symmetry test |
|---|---|---|---|---|
| **Piz Daint** | Fortran: OpenMP (CPU) | `fortran` (32) | `nl` (32) | `ad` (32) |
|  | Fortran: OpenACC (GPU) | `gpu-scc-k-caching` (128) | - | - |
|  | Fortran: Loki (GPU) | `loki-scc-cuf-hoist` (128) | `nl-loki-scc-hoist` (64) | - |
|  | C: CUDA (GPU) | `cuda-k-caching` (128) | - | - |
| **MeluXina** | Fortran: OpenMP (CPU) | `fortran` (32) | `nl` (32) | - |
|  | Fortran: OpenACC (GPU) | `gpu-scc-k-caching` (128) | - | - |
|  | Fortran: Loki (GPU) | `loki-scc-cuf-hoist` (128) | `nl-loki-scc-hoist` (128) | - |
|  | C: CUDA (GPU) | `cuda-k-caching` (128) | - | - |
| **LUMI** | Fortran: OpenMP (CPU) | `fortran` (32) | `nl` (32) | `ad` (32) |
|  | Fortran: OpenACC (GPU) | `gpu-scc-k-caching` (256) | - | - |
|  | Fortran: Loki (GPU) | `loki-scc-hoist` (256) | `nl-loki-scc-hoist` (256) | - |
|  | C: HIP (GPU) | `hip-k-caching` (64) | - | - |

**Table 3.** For each reference implementation of the CLOUDSC and CLOUDSC2 dwarfs, the table reports the string identifying the specific variant considered in the performance analysis on Piz Daint, MeluXina and LUMI. The corresponding NPROMA is provided within parentheses. Those implementations which are either not available or not working are marked with a dash.

node for Piz Daint, MeluXina and LUMI, with the compute and memory resources being evenly distributed across the NUMA domains. Note that the compute nodes of the GPU partition of LUMI have the low-noise mode activated, which reserves one core per NUMA domain to the operating system, so that only 7 out of 8 cores are available to the jobs. Moreover, each MI250X GPU is split into two virtual GPUs (vGPUs), with each vGPU assigned to a different

NUMA domain.

Figures 3-5 visualize the execution times for CLOUDSC (left column), CLOUDSC2NL (center column) and the symmetry test for CLOUDSC2TL and CLOUDSCAD (right column) for Piz Daint, MeluXina and LUMI, respectively. All performance numbers refer to a grid size of 65536 columns, with each column featuring 137 vertical levels. In each figure, execution times are provided for simulations running either entirely in double precision (FP64; top row) or in

single precision (FP32; bottom row). Within each panel, the plotted bars reflect the execution time of the various codes, with a missing bar indicating the corresponding code (non-GT4Py) is either not available or not working properly. Specifically,

    – the Fortran version of CLOUDSC2AD can only run on a single OpenMP thread on MeluXina (the issue is still under investigation);

– a native GPU-enabled version of CLOUDSC using 32-bit floating point arithmetic does not exist at the time of writing, and no CUDA/HIP implementations are available for CLOUDSC2;



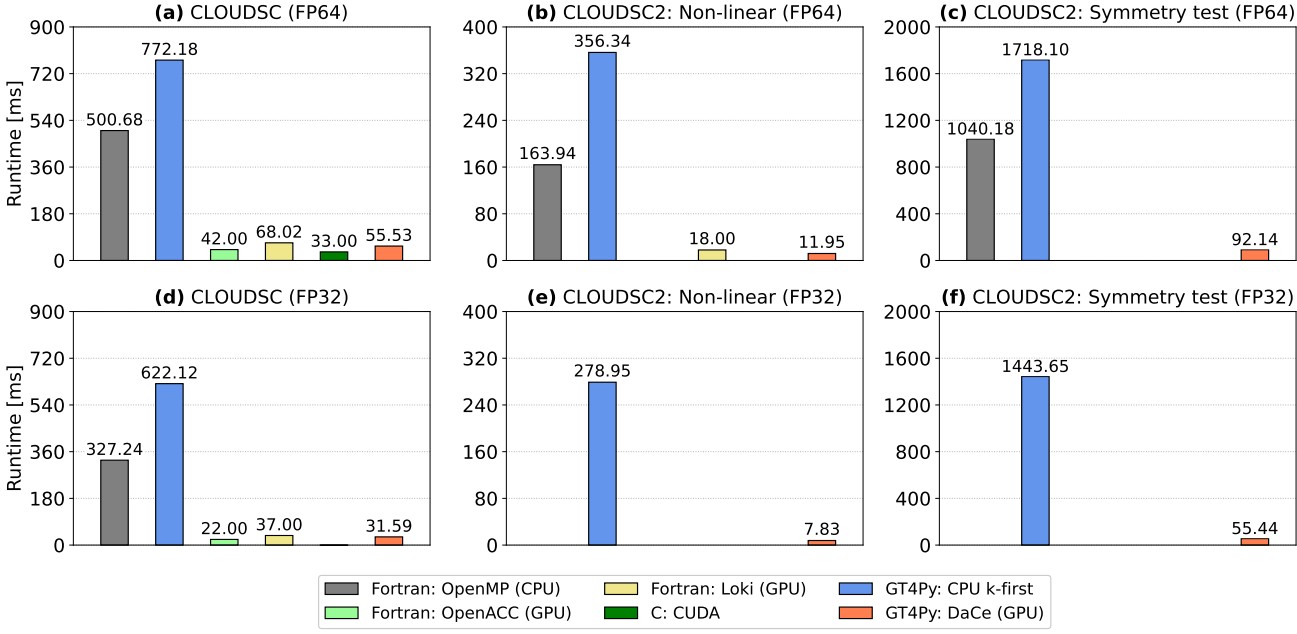

**Figure 3.** Execution time on a single NUMA domain of a hybrid node of the Piz Daint supercomputer for CLOUDSC (left column), CLOUDSC2NL (center column) and the symmetry test for CLOUDSC2TL and CLOUDSC2AD (right column) using either double precision (top row) or single precision (bottom row) floating point arithmetic. The computational domain consists of 65536 columns and 137 vertical levels. Displayed are the multi-threaded Fortran baseline using OpenMP (grey); two GPU-accelerated Fortran implementations, either using OpenACC directives (lime) or the source-to-source translation tool Loki (yellow); an optimized CUDA C version (green); and the GT4Py rewrite, either using the GridTools C++ CPU backend with k-first data ordering (blue) or the DaCe GPU backend (orange). All numbers should be interpreted as an average over 50 realizations. The panels only show the code versions available and validating at the time of writing.

- all Fortran-based implementations of the three formulations of CLOUDSC2 can only use double precision computations;

- a Loki version of CLOUDSC2TL and CLOUDSC2AD is not available at the time of writing.

Notably, we find the GT4Py rewrite of both CLOUDSC and CLOUDSC2 to be very robust, as the codes execute on every CPU and GPU architecture included in the study, and can always employ either double or single precision floating point arithmetic. With GT4Py, changing the backend with the respective target architecture, or changing the precision of computations, is as easy as setting a namelist parameter. Moreover, at the time of writing the GT4Py implementations of the more complex tangent-linear and adjoint formulations of CLOUDSC2 were the first codes

enabling GPU execution, again both in double or single precision.



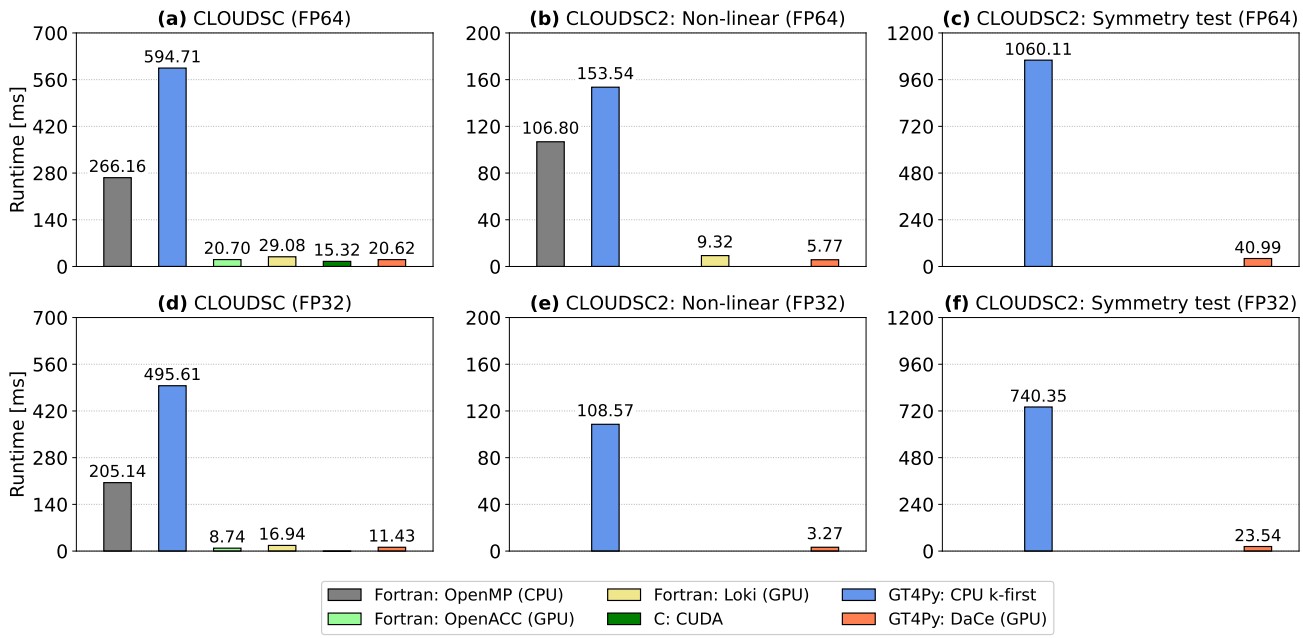

**Figure 4.** As Fig. 3 but for the MeluXina supercomputer.

The performance of the high-level Python with GT4Py compares well against Fortran with OpenACC. The runtimes for GT4Py with its DaCe backend versus OpenACC are similar on Piz Daint, MeluXina and LUMI. One outlier is the double precision result on LUMI, for which the OpenACC code appears relatively slow. We suppose this behaviour is associated with the insufficient OpenACC support for the HPE Cray compiler. Only the HPE Cray compiler implements GPU offloading capabilities for OpenACC directives on AMD GPUs, meaning that Fortran OpenACC codes require an HPE Cray platform to run on AMD GPUs. In contrast, GT4Py relies on the HIPCC compiler driver developed by AMD to compile device code for AMD accelerators, and this guarantees a proper functioning irrespective of the machine vendor. We further note that the DaCe backend of GT4Py executes roughly two times faster on MeluXina's NVIDIA A100 GPUs than on LUMI's AMD Instinct MI250X GPUs. As mentioned above, from a software perspective, each physical GPU module on LUMI is considered as two virtual GPUs, so that the code is actually executed on half of a physical GPU card. We can therefore speculate that if using both dies of an AMD Instinct MI250X GPU performance would be on par with the NVIDIA A100 GPU.

Another interesting result is that both CLOUDSC-GT4Py and CLOUDSC2-GT4Py are consistently faster than the implementations generated with Loki. Loki allows to build bespoke transformation recipes to apply changes to programming models and coding styles in an automated fashion. Therefore, GPU-enabled code can be produced





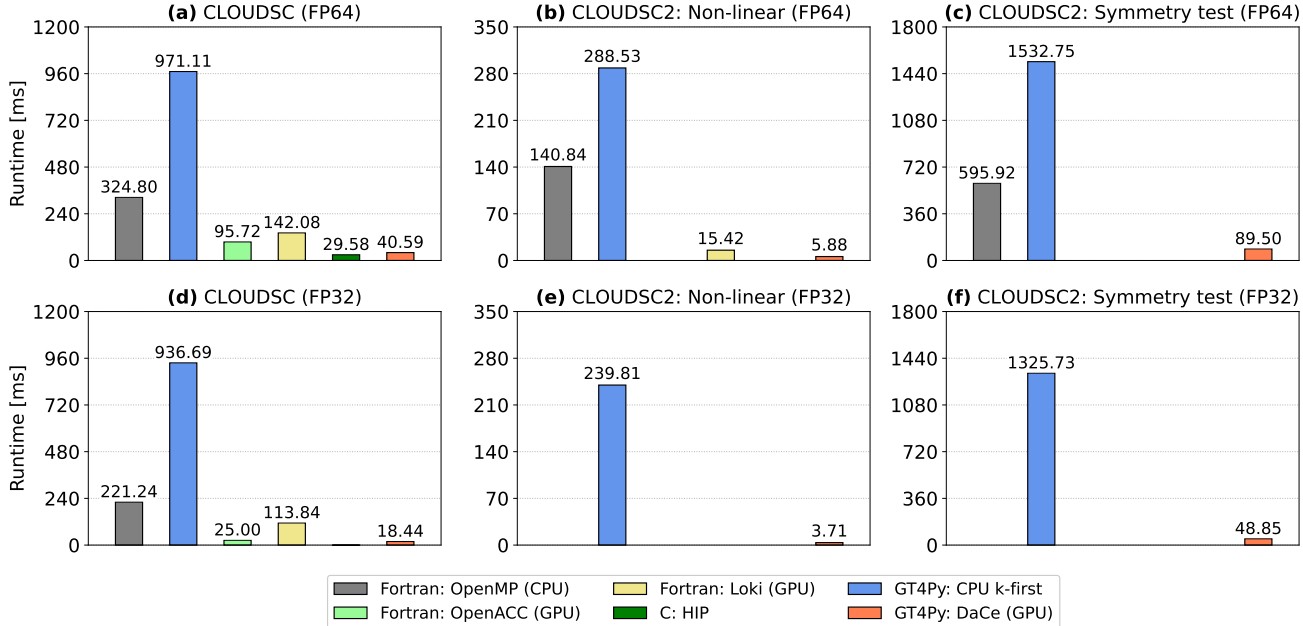

**Figure 5.** As Fig. 3 but for the LUMI supercomputer.

starting from the original Fortran by e.g., automatically adding OpenACC directives. However, because not all optimizations are yet encoded in the transformations, the Loki-generated device code cannot achieve optimal performance. Notwithstanding, source-to-source translators such as Loki are of high relevance for enabling GPU execution with large legacy Fortran code bases.

As used in this paper, GT4Py cannot yet attain the performance achieved by manually optimized native implementations with either Fortran on CPU or CUDA/HIP on GPU. Multi-threaded Fortran can be up to three times faster than the GridTools CPU backend of GT4Py using the k-first (C-like) memory layout, while the DaCe GPU backend of GT4Py can be up to a factor of two slower than CUDA/HIP. On the one hand, so far the development of GT4Py has been focused on GPU execution (see e.g. Dahm et al. (2023)), because this will be the dominant

hardware for time-critical applications in the years to come. On the other hand, we stress that the k-caching CUDA and HIP variants of CLOUDSC were semi-automatically generated by performance engineering experts, starting from an automatic Fortran-to-C transpilation of the SCC variants and manually applying additional optimizations that require knowledge about the specific compute patterns in the application. This process is not scalable to the full weather model and not a sustainable code adaptation method. In contrast, no significant performance engineering

has been applied yet with CLOUDSC-GT4Py and CLOUDSC2-GT4Py.



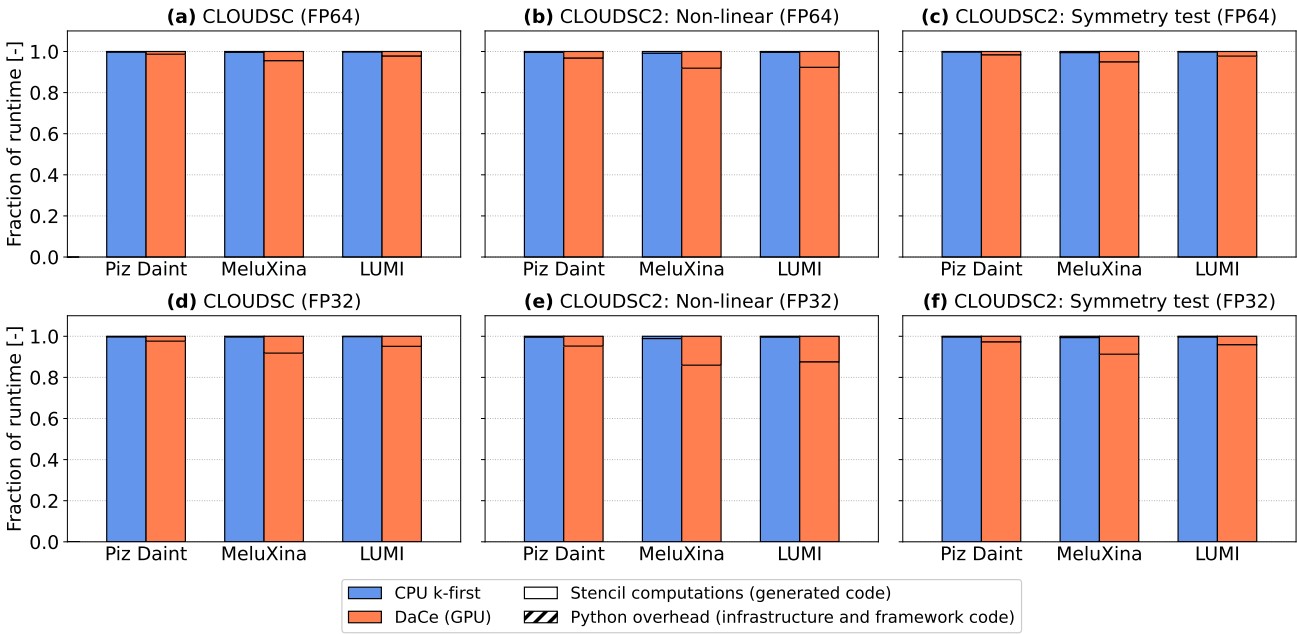

**Figure 6.** For the GT4Py rewrites of CLOUDSC (left column), CLOUDSC2NL (center column) and the symmetry test for CLOUDSC2TL and CLOUDSC2AD (right column), fraction of the total execution time spent within the stencil computations (full bars) and the Python side of the application (hatched bars) on Piz Daint, MeluXina and LUMI. Results are shown for the GridTools C++ CPU backend with k-first data ordering (blue) and the DaCe GPU backend (orange), either using double precision (top row) or single precision (bottom row) floating point arithmetic.

To rule out the possibility that the performance gap between the Python DSL and lower-level codes is associated with overhead originating from Python, Fig. 6 displays the fraction of runtime spent within the stencil code generated by GT4Py and the high-level Python code of the application (infrastructure and framework code; see Section 4). Across the three supercomputers, the Python overhead decreases as (i) the complexity and length of computations increase, (ii) the peak throughput and bandwidth delivered by the hardware underneath decrease, and (iii) the floating point precision increases. On average, the Python overhead accounts for 5.4% of the total runtime on GPU and 0.4% on CPU.

Finally, we observe a significant sensitivity of the GPU performance with respect to the thread block size[12]: for values smaller than 128, performance is degraded across all implementations, with the gap between CUDA/HIP and GT4Py+DaCe being smaller. This shows that some tuning and toolchain optimizations can be performed to improve performance with the DSL approach.

---

[12]In the Fortran code, the thread block size corresponds to the NPROMA.



## 6   Conclusions

The CLOUDSC and CLOUDSC2 cloud microphysics schemes of the IFS at ECMWF have served as demonstrators to show the benefits of a high-level domain-specific implementation approach to physical parametrizations. We presented two Python implementations based on the GT4Py framework, in which the scientific code is insulated from hardware-specific aspects. The resulting application provides a productive user interface with enhanced readability and maintainability, and can run efficiently and in a very robust manner across a wide spectrum of compute architectures/systems. The approach can be powerful in the light of the increasingly complex HPC technology landscape, where general-purpose CPUs are increasingly complemented by domain-specific architectures (DSAs) such as GPUs, Tensor Processor Units (TPUs), Field Programmable Gate Arrays (FPGAs), and Application-Specific Integrated Circuits (ASICS). In addition to the CLOUDSC scheme used in the IFS forecast model, we have presented results with the GT4Py rewrites of the nonlinear, tangent-linear and adjoint formulations of CLOUDSC2 used in data assimilation.

In both CLOUDSC-GT4Py and CLOUDSC2-GT4Py, the stencil kernels are encapsulated within model components sharing a minimal and clear interface. By avoiding any assumption on the host model, the interface aims to provide interoperable and plug-and-play physical packages, which can be transferred more easily between different modeling systems with virtually no performance penalty.

We carried out a comprehensive study to assess the portability of the Python codes across three major supercomputers, differing in terms of the vendor, the node architecture, the software stack and the compiler suites. We showed that the GPU performance of GT4Py codes are competitive against optimized Fortran OpenACC implementations and perform particularly well when compared to the available codes generated with the Loki source-to-source translation tool. Low-level implementations written either in Fortran for CPUs or CUDA/HIP for GPUs, with additional optimizations that possibly require knowledge about the specific compute patterns, can provide better performance, but are extremely challenging to create and maintain for entire models. The CPU performance of GT4Py is currently suboptimal, but this is an expected result given the focus on GPUs in the DSL development so far and we clearly expect this to improve significantly with upcoming and future GT4Py versions.

The presented results, based on a representative physical parametrization and considering tangent-linear and adjoint versions, add to the notion that weather and climate model codes can execute significantly faster on GPUs



(Fuhrer et al., 2018), and the number of HPC systems with accelerators is steadily increasing[13]. Therefore, we envision

that CPUs will be increasingly relegated to tasks that are not time-critical.

The current study supports our ongoing efforts and plans to port other physical parametrizations to Python with GT4Py. However, we note that GT4Py has been originally devised to express the computational motifs of dynamical cores based on grid-point methods, so not all patterns found in the parametrizations are natively supported by the DSL. These cases may be addressed by new features added to GT4Py or by resorting to other Python libraries to

generate fast machine code.

*Code and data availability.* The source codes for `ifs-physics-common` (Ubbiali et al., 2024b, https://github.com/stubbiali/ ifs-physics-common), CLOUDSC-GT4Py (Ubbiali et al., 2024c, https://github.com/stubbiali/gt4py-dwarf-p-cloudsc) and CLOUDSC2-GT4Py (Ubbiali et al., 2024d, https://github.com/stubbiali/gt4py-dwarf-p-cloudsc2-tl-ad), as well as the data and scripts to produce all the figures of the paper (Ubbiali et al., 2024a, https://github.com/stubbiali/cloudsc-paper), are

available on Github and archived on Zenodo.

*Author contributions.* SU ported the CLOUDSC and CLOUDSC2 dwarfs to Python using GT4Py and ran all the numerical experiments presented in the paper, under the supervision of CK and HW. SU further contributed to the development of the infrastructure code illustrated in Section 4, under the supervision of CS, LS and TCS. MS made relevant contributions to the Fortran and C reference implementations of the ECMWF microphysics schemes. SU and CK wrote the paper, with feedback

from all co-authors.

*Competing interests.* The authors declare that they have no conflict of interest.

*Acknowledgements.* This study was conducted as part of the Platform for Advanced Scientific Computing (PASC) funded project KILOS ("Kilometer-scale non-hydrostatic global weather forecasting with IFS-FVM"), which also provided us with computing resources on the Piz Daint supercomputer at CSCS. We acknowledge EuroHPC JU for awarding the project ID

---

[13]In the 62nd edition of the TOP500 list published in November 2023, 186 out of the 500 most powerful supercomputers in the world use graphics accelerator technology (https://www.top500.org/lists/top500/2023/11/highs/).





200177 access to the MeluXina supercomputer at LuxConnect and the project ID 465000527 access to the LUMI system at CSC, and thank Thomas Geenen and Nils Wedi from Destination Earth for their help. We are grateful to Michael Lange and Balthasar Reuter for discussions and support regarding IFS codes.



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
