# Peer review of "Exploring a high-level programming model for the NWP domain using ECMWF microphysics schemes"

_Geoscientific Model Development, 2024_

## Referee Comment (RC2)

Review of https://doi.org/10.5194/gmd-2024-92.

**Title of the Manuscript:** Exploring a high-level programming model for the NWP domain using ECMWF microphysics schemes

**Manuscript ID:** gmd-2024-92

**Date:** 14/6/2024
* * *
**Summary of the Manuscript**

The paper describes how to use the GT4Py library to implement a representative physical parametrization scheme and its related tangent-linear and adjoint algorithms from the IFS. The main objectives of the study are to demonstrate the correctness and performance-portability of Python rewrites with GT4Py against the reference Fortran code and various ported variants. The paper is part of a larger effort to port weather and climate models to Python with GT4Py, with a particular focus on the IFS prognostic cloud microphysics scheme.

The methods involve benchmarking prototype codes on three different HPCs with diverse hardware and software configurations to ensure robust execution and competitive performance. This includes comparing single and double precision variants. The results show good portability and reasonable performance of the Python rewrites with GT4Py across all tested scenarios.

Overall this paper is a useful contribution towards efforts to achieve performance portability and future-proofing of our model codes. Although I have raised some questions about the scientific correctness of the transferred code, these issues are not essential to the main goal of the paper – which is to explain the software aspects.

**GMD aspects for consideration**

**Does the paper address relevant scientific modelling questions within the scope of GMD?**
Yes.  This is a development and technical paper detailing technical implementation of a Domain Specific Language.

**Does the paper present a model, advances in modelling science, or a modelling protocol that is suitable for addressing relevant scientific questions within the scope of EGU?**
Yes. The application here would lead to improved modelling of the atmosphere and physical parametrizations in particular.

**Does the paper present novel concepts, ideas, tools, or data?**
Yes.  The GT4Py tooling is a novel approach to deal with performance portability.

**Does the paper represent a sufficiently substantial advance in modelling science?**
Yes.  The capability demonstrated could potentially lead to more performant and portable codes.

**Are the methods and assumptions valid and clearly outlined?**
Yes, though some suggestions for improvements have been provided below.

**Are the results sufficient to support the interpretations and conclusions?**
Yes, though some suggestions for improvements have been provided below.

**Is the description sufficiently complete and precise to allow their reproduction by fellow scientists (traceability of results)?**
Yes. The codes that have been used to produce these results are accessible and their location has been provided.

**Do the authors give proper credit to related work and clearly indicate their own new/original contribution?**
Yes, I believe so.

**Does the title clearly reflect the contents of the paper?**
Yes

**Does the abstract provide a concise and complete summary?**
Yes

**Is the overall presentation well structured and clear?**
Yes

**Is the language fluent and precise?**
Yes

**Are mathematical formulae, symbols, abbreviations, and units correctly defined and used?**
Yes

**Should any parts of the paper (text, formulae, figures, tables) be clarified, reduced, combined, or eliminated?**
I have suggested some additional comment would be useful for the algorithms.

**Are the number and quality of references appropriate?**
Yes.

**Is the amount and quality of supplementary material appropriate?**
Yes.

**General Comments**

1. Line 15: 'reduced precision' is referred to throughout the manuscript, but this is specifically 32-bit IEEE (single precision). As GPUs in particular can (and most Fortran compilers in general cannot currently) exploit other floating point models, e.g. 16-bit IEEE or 32-bit bfloat, I think it would be better to be specific about which precision. As a general question I wonder if you are well placed to explore these other floating-point models with GT4Py?

2. Line 96: You discuss a motivation for using the particular CLOUDSC schemes is that they are representative of the computational patterns in physical parametrizations. It would be good to be explicit about what those patterns are. In contrast, on line 398, you talk about not all patterns in parametrizations being natively supported.

I would suggest part of this discussion should clarify that the patterns under consideration with CLOUDSC are based on the spatial gridded structure (and in the column), while other parametrizations may use 'pseudo-dimensions' such as number of spectral bands, land surface types, or even moments or bins for more complex microphysics schemes. These may then require additional computational motifs and looping structures to extract optimal performance on different hardware.

3. Line 139/Algorithm 1: I'm not entirely sure the details of the TaylorTest are necessary for this paper. However, if you wish to keep it in then please could you add some brief inline documentation (as you might with real code) as to what the algorithm is doing at each stage. It is quite a difficult algorithm to read without it.

4. Line 188/Figure 2: The example provided here is for the 2D horizontal Laplacian which uses a stencil accessing horizontal neighbouring columns, but no vertical neighbouring grid points. This is the opposite of the microphysics stencil which does not require access to horizontal neighbours, but does require vertical neighbours (due to sedimentation of hydrometeors). Since this paper is specifically applying the method to the latter, it would be best if the example related to that. At the very least, having some text alongside the discussion of Figure 2 to explain this would be helpful.

5. Line 285: I'm not sure why you use the 'allclose' method to determine if a particular tolerance has been met rather than calculating and reporting the error. That said, my (limited) understanding of the allclose method is that only one of rtol and atol is sufficient to pass the test. As a result, the other can be made arbitrarily small and still pass the test. E.g.

```
>>> x=1.0 + 0.99e-12
>>> y=1.0
>>> np.allclose(y, x, atol=1e-12, rtol=0)
True
>>> np.allclose(y, x, atol=0, rtol=1e-18)
False
>>> np.allclose(y, x, atol=1e-12, rtol=1e-18)
True
```

So how did you use this function to arrive at the quoted numbers? (Presumably one would want to use the first two methods demonstrated independently for each tolerance.)
I think it would be more informative to simply provide values of max(abs(x-y)) and max(abs(x-y)/abs(x))  (where x is the reference data and y is the rewrite).

6. There seems to be more to investigate here to ensure correctness. The previous comment would go some way to helping with this, but the fact that the relative tolerance remains the same, but the absolute tolerance is 2 orders of magnitude larger suggests that the evolution of the experiment has changed. This could be a symptom of perturbation error growth where a small change leads to a different branch of the code being followed. To add some insight into this, it would also be useful to add more detail about the scientific set up of the dwarf being tested - what are the initial conditions and

timestepping - or where is this described?

7. Line 291: Was machine epsilon to single precision used in these tests?

8. Line 355: Could you say how performance engineering would be done with GT4Py? Would this be in the 'Optimizations' step in Figure 2?

9. Line 397: As per comment 2 above, it would be good to expand on what other patterns might be needed that aren't natively supported.

**Specific Comments**

1. Line 17: 'has become' should be 'became'.
2. Line 56: References for the GungHo dynamical core can be found at https://doi.org/10.1002/qj.3501 and https://arxiv.org/abs/2402.13738
3. Line 103: 'slightly polished' could perhaps be a bit more informative.  What needed to be done? Was it purely cosmetic?
4. Line 141:  I think there is a missing $\delta$ in the first inner product.
5. Line 142: 'hearth' should be 'heart'.
6. Line 180: 'scientists is exposed' should be 'scientists are exposed'.
7. Line 237: 'it builds' should be 'does it build'.
8. Line 237: It's not clear to me what 'grid-aware' means in this context. Could you be specific?
9. Line 296: I don't think NPROMA has been defined.

---

## Author Comment (AC1)

**gmd-2024-92**

**Exploring a high-level programming model for the NWP domain using ECMWF microphysics schemes**

by S. Ubbiali et al.

We thank the three reviewers for their constructive comments, which help us to further improve the manuscript. Below we provide a one-to-one response to all points raised by the reviewers. The reviewers' comments are in gray italics and our replies in black roman.

**RC1 (posted by Anonymous Referee #1 on 25.06.2024)**

*As a developer who is using GT4Py to port parameterized physics, I am encouraged by these performance results as well as the portability across multiple GPU architectures. Overall, I think this is an excellent paper that highlights the potential of DSLs as a forward-looking development platform. I have several questions and comments.*

We are grateful to the Referee for the overall positive feedback and many valuable comments. We find it particularly rewarding that another GT4Py user appreciates our work. In the following, we address each of the points raised by the Referee individually.

1. *Line 191 : This line mentions that "can be differentiated for the vertical boundaries using the interval context manager". As a GT4Py user, it's clear what is being written, but given that "differentiated" has mathematical meanings, it may be better to reword this to avoid confusion.*

   The term "differentiated" is indeed overloaded. We will rephrase the sentence in the revised version of the manuscript to avoid any misinterpretation.

2. *List 1 and 2 : I realized later that the "Code and data availability" section lists the repositories that contain the codes in List 1 and 2. Originally, I had mistakenly searched the ECMWF-iFS Github site for the CLOUDSC and CLOUDSC2 dwarf codes and was wondering why I*

*couldn't find the codes from the list. One suggestion is to mention that the repos for the codes are mentioned later in the "Code and data availability" section.*

To help readers find the correct code repositories, in the revised manuscript we will add a reference to Ubbiali et al. (2024b-d) right in the Introduction.

3. *Line 296 : Can NPROMA be explained further?*

We thank the Referee for touching upon this aspect, which has been raised in other reviews as well. In the revised manuscript, we will shortly mention what NPROMA represents and refer the reader to available literature resources (e.g. Bauer et al., 2020; doi.org/10.21957/gdit22ulm, and Müller et al., 2019; doi.org/10.5194/gmd-12-4425-2019) for further details.

4. *Line 307 : To clarify, is the symmetry test timing the sum of the CLOUDSC2TL and CLOUDSCAD timings?*

Yes, within the symmetry test both CLOUDSC2TL and CLOUDSCAD are called and considered in the timings. Conversely, the computation of the column-wise inner products (L10 & L12 in Algorithm 2) and the following validation procedure (L13-23 in Algorithm 2) are switched off when measuring performance.

5. *Line 336 : I'm a bit confused on the virtual GPU explanation. Does this mean that when 1 MPI process is mapped to an MI250X, only half the GPU is executed?*

That is correct: a MI250X GPU consists of two Graphics Compute Dies (GCDs) connected via four AMD Infinity Fabric links but not sharing physical memory and therefore each MPI rank naturally maps to a single GCD of a MI250X GPU.

6. *Question: The Gridtools backend was mentioned as a GT4Py backend (and I think it enables GPU compute), but its results were not presented. Was it because it was slower than the Dace backend?*

Both the GridTools and the DaCe backends of GT4Py enable GPU computing on both NVIDIA and AMD GPUs, offering very similar performance on MeluXina and LUMI. However, we could not compile the CLOUDSC stencil using the GridTools GPU backend on Piz Daint,

presumably because of a bug in CUDA11. We therefore decided to show results for the DaCe GPU backend only, as they were available on all three machines. On the other hand, the GridTools CPU backend was found to be faster than the DaCe CPU backend in all tested scenarios, so that is why we only present performance numbers for GridTools CPU. This is aligned with all the other programming paradigms considered in this study, for which only the fastest variant is taken into account.

**RC2 (posted by Anonymous Referee #2 on 26.06.2024)**

*Overall this paper is a useful contribution towards efforts to achieve performance portability and future-proofing of our model codes. Although I have raised some questions about the scientific correctness of the transferred code, these issues are not essential to the main goal of the paper – which is to explain the software aspects.*

We thank the Referee for such a careful revision of the manuscript, leading to many constructive comments, valuable suggestions and useful corrections. We are pleased to read the Referee's appreciation of our work. In the following, we address each of the issues raised by the Referee individually.

**General Comments**

1. *Line 15: 'reduced precision' is referred to throughout the manuscript, but this is specifically 32-bit IEEE (single precision). As GPUs in particular can (and most Fortran compilers in general cannot currently) exploit other floating point models, e.g. 16-bit IEEE or 32-bit float, I think it would be better to be specific about which precision. As a general question I wonder if you are well placed to explore these other floating-point models with GT4Py?*

   We agree with the Referee that the term "reduced precision" may be too vague. In the revised manuscript, we will replace "reduced" with "single" and clarify that this refers to the 32-bit IEEE standard.

   Although GT4Py does only support 64-bit and 32-bit floats at the moment, it will be pretty straightforward to allow 16-bit floats in the future.

2. *Line 96: You discuss a motivation for using the particular CLOUDSC schemes is that they are representative of the computational patterns in physical parametrizations. It would be good to be explicit about what those patterns are. In contrast, on line 398, you talk about not all patterns in parametrizations being natively supported. I would suggest part of this discussion should clarify that the patterns under consideration with CLOUDSC are based on the spatial gridded structure (and in the column), while other parametrizations may use 'pseudo-dimensions' such as number of spectral bands, land surface types, or even moments or bins for more complex microphysics*

*schemes. These may then require additional computational motifs and looping structures to extract optimal performance on different hardware.*

We thank the Referee for touching upon this important point. At the time of writing, one important limitation is indeed that multi-dimensional arrays are not supported in a performant manner. While this is not a problem here, it could be for other scenarios (e.g., radiation). Other features GT4Py is currently missing and that could be useful for physics codes include single column abstraction, write (absolute) offsets, and vertical reductions. In the revised version of the article, we will provide further details on this topic in the last paragraph of Section 6.

3. *Line 139/Algorithm 1: I'm not entirely sure the details of the TaylorTest are necessary for this paper. However, if you wish to keep it in then please could you add some brief inline documentation (as you might with real code) as to what the algorithm is doing at each stage. It is quite a difficult algorithm to read without it.*

To the best of our knowledge, there does not exist any good reference online for the Taylor test, so we would like to keep Algorithm 1. However, we agree with the Referee that the details of the test are not essential for the main purposes of the paper, therefore we decided to move the Algorithm to the Appendix. We will also follow the suggestion of the Referee and add inline documentation to the algorithm.

4. *Line 188/Figure 2: The example provided here is for the 2D horizontal Laplacian which uses a stencil accessing horizontal neighbouring columns, but no vertical neighbouring grid points. This is the opposite of the microphysics stencil which does not require access to horizontal neighbours, but does require vertical neighbours (due to sedimentation of hydrometeors). Since this paper is specifically applying the method to the latter, it would be best if the example related to that. At the very least, having some text alongside the discussion of Figure 2 to explain this would be helpful.*

We agree with the Referee that the example shown in the code snippet of Fig. 2 does not feature the characteristic access patterns of the microphysics stencils. We will choose a more representative example in the revised manuscript. However, since the GT4Py internal workflow is described in the main body and Fig. 2 is supposed to only be a visual aid, we would avoid congesting the figure with additional text.

5. *Line 285: I'm not sure why you use the 'allclose' method to determine if a particular tolerance has been met rather than calculating and reporting the error. That said, my (limited) understanding of the allclose method is that only one of rtol and atol is sufficient to pass the test. As a result, the other can be made arbitrarily small and still pass the test. E.g.*

*>>> x=1.0 + 0.99e-12*

*>>> y=1.0*

*>>> np.allclose(y, x, atol=1e-12, rtol=0)*

*True*

*>>> np.allclose(y, x, atol=0, rtol=1e-18)*

*False*

*>>> np.allclose(y, x, atol=1e-12, rtol=1e-18)*

*True*

*So how did you use this function to arrive at the quoted numbers? (Presumably one would want to use the first two methods demonstrated independently for each tolerance.)*

*I think it would be more informative to simply provide values of max(abs(x-y)) and max(abs(x-y)/abs(x)) (where x is the reference data and y is the rewrite).*

In the Python community, it is pretty common to use the *isclose* function from Numpy to check whether two numbers $x$ (reference) and $y$ are close up to absolute tolerance *atol* and relative tolerance *rtol*: $isclose(x, y, atol, rtol) = abs(x - y) < atol + abs(x) * rtol$. So both tolerance values are employed by the function simultaneously, and the values we report in the text are the smallest ensuring that *isclose* returns *True* on all grid points for all output fields (i.e., further decreasing *atol* or *rtol* would make the validation fail). Hence, we do not think that providing the absolute and relative errors would be an added value for the paper.

6. *There seems to be more to investigate here to ensure correctness. The previous comment would go some way to helping with this, but the fact that the relative tolerance remains the same, but the absolute tolerance*

*is 2 orders of magnitude larger suggests that the evolution of the experiment has changed. This could be a symptom of perturbation error growth where a small change leads to a different branch of the code being followed. To add some insight into this, it would also be useful to add more detail about the scientific set up of the dwarf being tested - what are the initial conditions and time-stepping - or where is this described?*

Our answer to the previous point should explain why the absolute and relative tolerance can have different orders of magnitude (depending on the magnitude of $y$).

With respect to the Referee's comment about error growth and time-stepping, we would like to point out that the dwarf codes tested in the paper do not involve integration of a complete atmospheric model. The study validates the developed GT4Py versions by reproducing the results of the IFS Fortran microphysics schemes, which represent the established codes for operational weather forecasting. Hence, we ensure the correct execution of the GT4Py codes for CLOUDSC and CLOUDSC2 by direct comparison with the baseline Fortran implementation on the basis of identical input data to produce the same output.. The input represents real data that is serialized from the IFS and which is available on Zenodo (cf. "Code and data availability" section). With regard to CLOUDSC2TL and CLOUDSC2AD, any minimal error in the implementation would have made either the Taylor test or the symmetry tests fail. In addition, we further note that the GT4Py implementation of CLOUDSC has been tested extensively in the context of a full atmospheric model in the meanwhile. Altogether, we are very confident about the correctness of our GT4Py implementations.

7. *Line 291: Was machine epsilon to single precision used in these tests?*

We were actually using double precision machine epsilon. We are very grateful to the Referee for guessing such inconsistency. We will correct the text and we can anticipate that when using single precision epsilon, the symmetry test for the single precision GT4Py implementations succeeds.

8. *Line 355: Could you say how performance engineering would be done with GT4Py? Would this be in the 'Optimizations' step in Figure 2?*

The "Optimizations" step in Fig. 2 enclose all the optimization strategies carried out internally by the GT4Py library. On the user side, performance can be improved by, e.g., fusing statements & stencils, and pruning temporaries. We will mention these aspects in the revised manuscript.

9. *Line 397: As per comment 2 above, it would be good to expand on what other patterns might be needed that aren't natively supported.*

Please see our reply to comment 2.

**Specific Comments**

1. *Line 17: 'has become' should be 'became'.*

Thank you, we will adapt the text according to the Referee's suggestion.

2. *Line 56: References for the GungHo dynamical core can be found at https://doi.org/10.1002/qj.3501 and https://arxiv.org/abs/2402.13738.*

We thank the Referee for providing recent references to the GungHo dynamical core. We will update the text accordingly.

3. *Line 103: 'slightly polished' could perhaps be a bit more informative. What needed to be done? Was it purely cosmetic?*

The CLOUDSC & CLOUDSC2 dwarfs do not differ substantially from the corresponding original implementations run operationally at ECMWF. The cleaning-up mostly consisted in removing (i) all the IFS-specific infrastructure code (that is not necessary to run the dwarfs stand-alone), (ii) the calculation of budget diagnostics, and (iii) dead codes (which would not be executed anyway). We will briefly touch upon these points in the text.

4. *Line 141: I think there is a missing $\delta$ in the first inner product.*

We thank the Referee for spotting the error, we will correct the text.

5. *Line 142: 'hearth' should be 'heart'.*

We thank the Referee for spotting the typo, we will correct the text.

6. *Line 180: 'scientists is exposed' should be 'scientists are exposed'.*

We thank the Referee for spotting the typo, we will correct the text.

7. *Line 237: 'it builds' should be 'does it build'.*

Thank you, we will adapt the text according to the Referee's suggestion.

8. *Line 237: It's not clear to me what 'grid-aware' means in this context. Could you be specific?*

We will briefly expand on this aspect in the revised paper. For instance, this includes that all components are instantiated over a ComputationalGrid (cf. Listing 2) which collects information about the underlying (Cartesian) grid, e.g. grid spacings, index spaces.

9. *Line 296: I don't think NPROMA has been defined.*

We thank the Referee for touching upon this aspect, which has been raised in other reviews as well. In the revised manuscript, we will shortly mention what NPROMA represents and refer the reader to available literature resources (e.g., Bauer et al., 2020; doi.org/10.21957/gdit22ulm, and Müller et al., 2019; doi.org/10.5194/gmd-12-4425-2019) for further details.

**RC3 (posted by Anonymous Referee #3 on 24.07.2024)**

*This is an excellent paper expanding the use of domain specific languages (DSLs), and GT4Py specifically, for performance and productivity in numerical weather prediction. To my knowledge this is the first published work on a tangent-linear or adjoint model in GT4Py, and the results are very encouraging. The authors describe their methodology and development process well, which will aid others looking to reproduce this work and apply it to their own models. That said I do have some small questions and comments I would like to raise before publication:*

We would like to express our sincere gratitude to the Referee for reviewing our work so carefully. We are very pleased to read the Referee's general appreciation of this study, and we thank them for the many constructive and valuable comments that we address point-by-point in the following.

**Primary points/questions:**

1. *I don't think it is necessary to define the tangent linear or adjoint operators explicitly, and I'm also not certain that you need to explicitly define the Taylor test either.*

   To the best of our knowledge, there does not exist any good reference online for the Taylor test, so we would like to keep Algorithm 1. However, we agree with the Referee that the details of the test are not essential for the main purposes of the paper, therefore we have decided to move the Algorithm to the Appendix.

2. *Line 247: I would like to see more description of the infrastructure code around the stencil. What does compile_stencil look like? Presumably the parent DiagnosticComponent class specifies the __call__ method, which wraps array_call, but that would be nice to see explicitly instead of assuming from what is in the paper.*

   We are glad to read the Referee's interest in our infrastructure code. However, it is beyond the scope of the paper to describe the infrastructure code in detail, and we think that this pertains more to a technical documentation, rather than a scientific paper. We refer the Reviewer and any interested reader to the source code of ifs-physics-common (https://github.com/stubbiali/ifs-physics-common)

and a talk given by the lead author of the paper at the recent PASC24 conference (https://event.pasc24-conference.org/slots/msa212).

3. *Line 250: Similarly, the stencil collection decorator is ifs-specific, and I would appreciate more detail about what it does and how.*

Please refer to our reply to the previous point.

4. *Line 265: Why use a GT4Py backend for CPU but a DaCe backend for gpu?*

For each programming paradigm (either in Fortran, C or Python) we only show performance numbers for the fastest variant. Since the GridTools CPU backend is found to be faster than the DaCe CPU backend, we only take into account the former. On the other hand, the GridTools GPU and DaCe GPU backends offer very similar performance on MeluXina and LUMI. However, we could not compile the CLOUDSC stencil using the GridTools GPU backend on Piz Daint, presumably because of a bug in CUDA11. We therefore decided to show results for the DaCe GPU backend only, as they were available on all three machines.

5. *Line 345: Is the goal of the GT4Py or ECMWF teams to achieve the same performance as native Fortran and CUDA models, or is it to attain most of their performance alongside the benefits of portability and productivity?*

We aim for productivity and portability while achieving competitive performance on both GPU and CPU. Since the DSL can accommodate any specific low-level optimization, attaining the same performance as native Fortran and CUDA models is feasible and will be the target of our future efforts.

6. *Figures 3-5: I'm not convinced by the layout of these figures. Because there are fewer implementations of CLOUDSC2 (and none in 32-bit aside from GT4Py) it may be more natural to report these performance results in a table, or to remove the space for the missing data, especially panels e and f which look disconcertingly sparse. On the other hand this is a very striking way to draw attention to the fact that GT4Py gives you 64- and 32-bit versions of the model in one go, but if you want to emphasize that I would like to see it more explicitly highlighted in the text.*

We are glad to read that the Referee finds the layout of Figs. 3-5 a striking way to emphasize the enhanced portability and flexibility of GT4Py codes. This is indeed our goal, therefore we would prefer to keep the figures as they are. We will carefully review the text to see whether and where we could further highlight the benefits provided by GT4Py.

**Minor:**

1. *In your introduction is it worthwhile to discuss efforts to use tools like Numba or Cython to accelerate numerical models written in Python across various fields of science, such as Augier et al. (doi:10.1038/s41550-021-01342-y) or others?*

   We thank the Referee for the meaningful suggestion. However, we note that it is beyond the scope of the paper to discuss how to accelerate Python codes in general. We believe that the Introduction is already very comprehensive.

2. *Line 18: the authors describe Fortran's "functional programming style" which is slightly imprecise; while Fortran uses functions and subroutines, functional programming refers to a style of programming using only pure functions, so no values are updated in-place, which is not how Fortran operates.*

   We thank the Referee for this clarification, which we agree with. We will take it into consideration in the revision process.

3. *Line 177: It would be useful to acknowledge contributions from groups beyond the Allen Institute, since they have ceased their work on GT4Py.*

   Only major partners of ETH Zurich are mentioned; the only exception is the Allen Institute for Artificial Intelligence (AI2), for their significant contributions to GT4Py.Cartesian.

4. *Line 190: "GTScript abstracts spatial for-loops away" would be more accurate than stating it abstracts for-loops entirely.*

   Thank you, we will adapt the text according to the Referee's suggestion.

5. *Line 237: "Not only it builds upon Sympl, but it also extends it" should be "Not only does it build upon Sympl, but also extends it".*

Thank you, we will adapt the text following the Referee's suggestion.

6. *Figure 6: Because the relevant information is contained within the top ~10% of the plot it may be useful to change the y-axis to instead range from 0.8 to 1.0.*

Although we understand the point made by the Referee, we find the plots in Fig. 6 to be more informative if the full bars are shown, so that one can better appreciate the fraction of the total runtime spent on the Python side.

7. *Listing 1: Should foealfcu be "foealpha"?*

We thank the Referee for spotting the typo, we will correct the text.

**Other comments:**

1. *Line 323: The fact that the GT4Py implementations of the tangent-linear and adjoint formulations of CLOUDSC2 are the first to enable GPU execution at any precision is very cool and could be emphasized more heavily throughout the paper, in my opinion.*

We are grateful to the Referee for acknowledging and appreciating our work. However, we would like to point out that the primary purpose of the paper is to show the effectiveness of the DSL approach (and in particular GT4Py) for the NWP domain, with respect to productivity, portability, and GPU performance. We believe the robustness of the GT4Py approach, including with respect to the TL/AD codes and single precision, becomes very clear from the current version of the manuscript.

2. *Line 361: It might be worth mentioning that the Python overhead would still account for around 1% of CPU runtime even if the GT4Py CPU performance was on par with Fortran.*

We thank the Referee for this consideration, which we will take into account in the revised manuscript.

---

## Author Response (AR1)

**gmd-2024-92**

**Exploring a high-level programming model for the NWP domain using ECMWF microphysics schemes**

by S. Ubbiali et al.

We thank the three reviewers for their constructive comments, which helped us to further improve the manuscript. Below we provide a one-to-one response to all points raised by the reviewers. The reviewers' comments are in *gray italics*, our replies in black roman, and changes to the manuscript are highlighted in blue roman at the end of each reply. In case a comment did not lead to any modification in the text, this is stated in red roman. Additional modifications to the text, not driven by any specific reviewers' comment, are reported at the end of the document. Please note that line numbers in our replies always refer to the revised version of the manuscript.

**RC1 (posted by Anonymous Referee #1 on 25.06.2024)**

*As a developer who is using GT4Py to port parameterized physics, I am encouraged by these performance results as well as the portability across multiple GPU architectures. Overall, I think this is an excellent paper that highlights the potential of DSLs as a forward-looking development platform. I have several questions and comments.*

We are grateful to the Referee for the overall positive feedback and many valuable comments. We find it particularly rewarding that another GT4Py user appreciates our work. In the following, we address each of the points raised by the Referee individually.

1. *Line 191 : This line mentions that "can be differentiated for the vertical boundaries using the interval context manager". As a GT4Py user, it's clear what is being written, but given that "differentiated" has mathematical meanings, it may be better to reword this to avoid confusion.*

   The term "differentiated" is indeed overloaded, and we avoid its use in the revised manuscript.

   L194-195: Replace *"differentiated for the vertical boundaries"* with *"can be controlled with respect to the vertical index bounds"*.

2. *List 1 and 2 : I realized later that the "Code and data availability" section lists the repositories that contain the codes in List 1 and 2. Originally, I had mistakenly searched the ECMWF-iFS Github site for the CLOUDSC and CLOUDSC2 dwarf codes and was wondering why I couldn't find the codes from the list. One suggestion is to mention that the repos for the codes are mentioned later in the "Code and data availability" section.*

   To help readers find the correct code repositories, in the revised manuscript we cite Ubbiali et al. (2024b-d) right in the Introduction.

   - L73: Add a reference to Ubbiali et al. (2024c) and Ubbiali et al. (2024d).
   - L79-80: Add a reference to Ubbiali et al. (2024b).

3. *Line 296 : Can NPROMA be explained further?*

We thank the Referee for touching upon this aspect, which has been raised in other reviews as well. In the revised manuscript, we shortly describe what the NPROMA represents and refer the reader to available literature resources (e.g. Bauer et al., 2020; doi.org/10.21957/gdit22ulm, and Müller et al., 2019; doi.org/10.5194/gmd-12-4425-2019) for further details on the NPROMA slicing technique.

L299: Add footnote 13: *"NPROMA blocking is a cache optimization technique adopted in all Fortran codes considered in this paper. Given a two-dimensional array shaped (K∗M, N), this is re-arranged as a three-dimensional array shaped (K, M, N). Commonly, the leading dimension of the three-dimensional array is called "NPROMA", with K being the "NPROMA blocking factor". Here, we indicate K simply as "NPROMA" for the sake of brevity. For further discussion of the NPROMA blocking, we refer the reader to Müller et al. (2019) and Bauer et al. (2020)."*.

4. *Line 307 : To clarify, is the symmetry test timing the sum of the CLOUDSC2TL and CLOUDSCAD timings?*

Yes, within the symmetry test both CLOUDSC2TL and CLOUDSCAD are called and considered in the timings. Conversely, the computation of the column-wise inner products (L11-12 of Algorithm 2) and the following validation procedure (L13-23 of Algorithm 2) are switched off when measuring performance. We added a footnote in the article to clarify that validation is not performed when timing the symmetry test.

L315: Add footnote 14: *"When measuring the performance of the symmetry test, the validation procedure – corresponding to lines 11-23 of Algorithm A2 – is switched off."*.
Algorithm A2: Swap L10 with L11.

5. *Line 336 : I'm a bit confused on the virtual GPU explanation. Does this mean that when 1 MPI process is mapped to an MI250X, only half the GPU is executed?*

That is correct: a MI250X GPU consists of two Graphics Compute Dies (GCDs) connected via four AMD Infinity Fabric links but not sharing physical memory and therefore each MPI rank naturally maps to a single GCD of a MI250X GPU. We rephrased the sentence at L307-310 to include all this information.

L309-312: Rephrase *"Moreover, each MI250X GPU is split into two virtual GPUs (vGPUs), with each vGPU assigned to a different NUMA domain."* as *"Moreover, we highlight that each MI250X GPU consists of two Graphics Compute Dies (GCDs) connected via four AMD Infinity Fabric links but not sharing physical memory. From a software perspective, each compute node of LUMI is equipped with 8 virtual GPUs (vGPUs), with each vGPU corresponding to a single GCD and assigned to a different NUMA domain."*.

6. *Question: The Gridtools backend was mentioned as a GT4Py backend (and I think it enables GPU compute), but its results were not presented. Was it because it was slower than the Dace backend?*

Both the GridTools and the DaCe backends of GT4Py enable GPU computing on both NVIDIA and AMD GPUs, offering very similar performance on MeluXina and LUMI. However, we could not compile the CLOUDSC stencil using the GridTools GPU backend on Piz Daint, presumably because of a bug in CUDA11. We therefore decided to show results for the DaCe GPU backend only, as they were available on all three machines. On the other hand, the GridTools CPU backend was found to be faster than the DaCe CPU backend in all tested scenarios, so that is why we only present performance numbers for GridTools CPU. This is aligned with all the other programming paradigms considered in this study, for which only the fastest variant is taken into account, as stated at L297-301.

L299-301: Add *"Similarly, for all Python implementations we consider only the most performant backends of GT4Py: the GridTools C++ CPU backend with k-first memory layout, and the DaCe GPU backend."*.

**RC2 (posted by Anonymous Referee #2 on 26.06.2024)**

*Overall this paper is a useful contribution towards efforts to achieve performance portability and future-proofing of our model codes. Although I have raised some questions about the scientific correctness of the transferred code, these issues are not essential to the main goal of the paper – which is to explain the software aspects.*

We thank the Referee for such a careful revision of the manuscript, leading to many constructive comments, valuable suggestions and useful corrections. We are pleased to read the Referee's appreciation of our work. In the following, we address each of the issues raised by the Referee individually.

**General Comments**

1. *Line 15: 'reduced precision' is referred to throughout the manuscript, but this is specifically 32-bit IEEE (single precision). As GPUs in particular can (and most Fortran compilers in general cannot currently) exploit other floating point models, e.g. 16-bit IEEE or 32-bit float, I think it would be better to be specific about which precision. As a general question I wonder if you are well placed to explore these other floating-point models with GT4Py?*

   We agree with the Referee that the term "reduced precision" may be too vague. In the revised manuscript, we use the term "single" in place of "reduced" in the abstract, while in Section 5 we now clarify that "double" refers to the 64-bit IEEE format and "single" corresponds to the 32-bit IEEE format.

   Although GT4Py does only support 64-bit and 32-bit floats at the moment, it will be pretty straightforward to allow 16-bit floats in the future.

   - L15: Replace *"reduced"* with *"single"*.
   - L317: Replace "*FP64*" with *"corresponding to the 64-bit IEEE format and denoted as FP64"*.
   - L317-318: Replace *"FP32"* with *"corresponding to the 32-bit IEEE format and denoted as FP32"*.

2. *Line 96: You discuss a motivation for using the particular CLOUDSC schemes is that they are representative of the computational patterns in physical parametrizations. It would be good to be explicit about what those patterns are. In contrast, on line 398, you talk about not all patterns in parametrizations being natively supported. I would suggest part of this discussion should clarify that the patterns under consideration with CLOUDSC are based on the spatial gridded structure (and in the column), while other parametrizations may use 'pseudo-dimensions' such as number of spectral bands, land surface types, or even moments or bins for more complex microphysics schemes. These may then require additional computational motifs and looping structures to extract optimal performance on different hardware.*

We thank the Referee for touching upon this important point. At the time of writing, one important limitation is indeed that multi-dimensional arrays are not supported in a performant manner. While this is not a problem here, it could be for other scenarios (e.g., radiation). Secondary features GT4Py is currently missing and that could be useful for physics codes include single column abstractions, write (absolute) offsets, and global reductions. In the revised version of the article, we mention the major current limitations of GT4Py in the last paragraph of Section 6.

L410-413: Replace *"These cases may be addressed by new features added to GT4Py or by resorting to other Python libraries to generate fast machine code."* with *"In this respect, current limitations of the DSL exist for higher-dimensional fields (e.g., arrays storing a tensor at each grid point), but again we expect these to be fully supported for HPC with future extensions of GT4Py. In addition, Python offers a variety of alternative libraries that can be employed in conjunction with GT4Py to generate fast machine code."*.

3. *Line 139/Algorithm 1: I'm not entirely sure the details of the TaylorTest are necessary for this paper. However, if you wish to keep it in then please could you add some brief inline documentation (as you might with real code) as to what the algorithm is doing at each stage. It is quite a difficult algorithm to read without it.*

To the best of our knowledge, there does not exist any good reference online for the Taylor test and the symmetry test, so we would like to keep both Algorithms 1 and 2. However, we agree with the Referee that

the details of the tests are not essential for the main purposes of the paper, therefore we decided to move both Algorithms to the new Appendix A titled "Algorithmic description of the Taylor test and symmetry test for CLOUDSC2". Since detailed understanding of the Algorithms is not necessary to grasp the essence of the article, we eventually decided not to add any more inline comments, so as not to make the pseudo-codes even longer and more visually cumbersome.

- Move Algorithm 1 and 2 to the new Appendix A titled *"Algorithmic description of the Taylor test and symmetry test for CLOUDSC2"*.
- L140-141: Add *"(Appendix A)"*.
- L146: Add *"in Appendix A"*.
- L415-417: Add *"In Section 2.2, we briefly described the aim and functioning of the Taylor test and the symmetry test for CLOUDSC2. Here, we detail the logical steps performed by the two tests with the help of pseudo-codes encapsulated in Algorithms A1 and A2."*.

4. *Line 188/Figure 2: The example provided here is for the 2D horizontal Laplacian which uses a stencil accessing horizontal neighbouring columns, but no vertical neighbouring grid points. This is the opposite of the microphysics stencil which does not require access to horizontal neighbours, but does require vertical neighbours (due to sedimentation of hydrometeors). Since this paper is specifically applying the method to the latter, it would be best if the example related to that. At the very least, having some text alongside the discussion of Figure 2 to explain this would be helpful.*

We agree with the Referee that the example shown in the code snippet of Fig. 2 does not feature the characteristic access patterns of the microphysics stencils. As a more representative example, we now employ the three-dimensional Laplacian operator. However, since the GT4Py internal workflow is described in the main body and Fig. 2 is supposed to only be a visual aid, we avoid congesting the figure with additional text.

- Figure 2: Modify the code snippet in the upper part of the diagram.
- Caption of Figure 2: Replace *"horizontal"* with *"three-dimensional"*.

5. *Line 285: I'm not sure why you use the 'allclose' method to determine if a particular tolerance has been met rather than calculating and reporting the error. That said, my (limited) understanding of the allclose method is that only one of rtol and atol is sufficient to pass the test. As a result, the other can be made arbitrarily small and still pass the test. E.g.*

   *>>> x=1.0 + 0.99e-12*

   *>>> y=1.0*

   *>>> np.allclose(y, x, atol=1e-12, rtol=0)*

   *True*

   *>>> np.allclose(y, x, atol=0, rtol=1e-18)*

   *False*

   *>>> np.allclose(y, x, atol=1e-12, rtol=1e-18)*

   *True*

   *So how did you use this function to arrive at the quoted numbers? (Presumably one would want to use the first two methods demonstrated independently for each tolerance.)*

   *I think it would be more informative to simply provide values of max(abs(x-y)) and max(abs(x-y)/abs(x)) (where x is the reference data and y is the rewrite).*

   In the Python community, it is pretty common to use the *isclose* function from Numpy to check whether two numbers $x$ (reference) and $y$ are close up to absolute tolerance *atol* and relative tolerance *rtol*: $isclose(x, y, atol, rtol) = abs(x - y) < atol + abs(x) * rtol$. So both tolerance values are employed by the function simultaneously, and the values we report in the text are the smallest ensuring that *isclose* returns *True* on all grid points for all output fields (i.e., further decreasing *atol* or *rtol* would make the validation fail). Hence, we do not think that providing the absolute and relative errors would be an added value for the paper.

   No changes have been made to the text.

6. *There seems to be more to investigate here to ensure correctness. The previous comment would go some way to helping with this, but the fact that the relative tolerance remains the same, but the absolute tolerance is 2 orders of magnitude larger suggests that the evolution of the experiment has changed. This could be a symptom of perturbation error growth where a small change leads to a different branch of the code being followed. To add some insight into this, it would also be useful to add more detail about the scientific set up of the dwarf being tested - what are the initial conditions and time-stepping - or where is this described?*

Our answer to the previous point should explain why the absolute and relative tolerance can have different orders of magnitude (depending on the magnitude of $y$).

With respect to the Referee's comment about error growth and time-stepping, we would like to point out that the dwarf codes tested in the paper do not involve integration of a complete atmospheric model. The study validates the developed GT4Py versions by reproducing the results of the IFS Fortran microphysics schemes, which represent the established codes for operational weather forecasting. Hence, we ensure the correct execution of the GT4Py codes for CLOUDSC and CLOUDSC2 by direct comparison with the baseline Fortran implementation on the basis of identical input data to produce the same output. The input represents real data that is serialized from the IFS and which is available on Zenodo (cf. "Code and data availability" section). With regard to CLOUDSC2TL and CLOUDSC2AD, any minimal error in the implementation would have made either the Taylor test or the symmetry tests fail. In addition, we further note that the GT4Py implementation of CLOUDSC has been tested extensively in the context of a full atmospheric model in the meanwhile. Altogether, we are very confident about the correctness of our GT4Py implementations.

No changes have been made to the text.

7. *Line 291: Was machine epsilon to single precision used in these tests?*

We were actually using double precision machine epsilon. We are very grateful to the Referee for guessing such inconsistency. We corrected the text, highlighting that when using the appropriate machine epsilon,

the symmetry test for the GT4Py implementations succeeds both with single and double precision, both on CPU and GPU.

L293-295: Replace *"However, the conditions of both tests are not satisfied when using single precision. This is not surprising, since both tests are highly sensitive to round-off errors. Nevertheless, performance numbers for the execution of the algorithms were taken."* with *"In this respect, we emphasize that the GT4Py implementations satisfy the conditions of both tests on all considered computing architectures, regardless of whether double or single precision is employed."*.

8. *Line 355: Could you say how performance engineering would be done with GT4Py? Would this be in the 'Optimizations' step in Figure 2?*

The "Optimizations" step in Fig. 2 enclose all the optimization strategies carried out internally by the GT4Py library. On the user side, performance can be improved by, e.g., fusing statements & stencils, and pruning temporaries. We briefly mention these aspects in the revised manuscript.

L363: Add *"(by, e.g., loop fusing and reducing the number of temporary fields)"*.

9. *Line 397: As per comment 2 above, it would be good to expand on what other patterns might be needed that aren't natively supported.*

Please see our reply to comment 2.

No changes have been made to the text.

**Specific Comments**

1. *Line 17: 'has become' should be 'became'.*

Thank you, we adapted the text according to the Referee's suggestion.

L17: Replace *"has become"* with *"became"*.

*2. Line 56: References for the GungHo dynamical core can be found at https://doi.org/10.1002/qj.3501 and https://arxiv.org/abs/2402.13738.*

We thank the Referee for providing recent references to the GungHo dynamical core. We updated the text accordingly.

- L56: Add references to Melvin et al. (2019; https://doi.org/10.1002/qj.3501) and Melvin et al. (2024; https://arxiv.org/abs/2402.13738).
- L530-532: Add bibliography entry for Melvin et al. (2019).
- L533-535: Add bibliography entry for Melvin et al. (2024).

*3. Line 103: 'slightly polished' could perhaps be a bit more informative. What needed to be done? Was it purely cosmetic?*

The CLOUDSC & CLOUDSC2 dwarfs do not differ substantially from the corresponding original implementations run operationally at ECMWF. The cleaning-up mostly consisted in removing (i) all the IFS-specific infrastructure code (that is not necessary to run the dwarfs stand-alone), (ii) the calculation of budget diagnostics, and (iii) dead codes (which would not be executed anyway). We added this information in a footnote.

L105: Add footnote 3: *"Compared to the original implementations run operationally at ECMWF, the CLOUDSC & CLOUDSC2 dwarf codes do not include (i) all the IFS-specific infrastructure code, (ii) the calculation of budget diagnostics, and (iii) dead code paths."*.

*4. Line 141: I think there is a missing $\delta$ in the first inner product.*

We thank the Referee for spotting the error, we corrected the text.

L145, Eq. 4: Add missing $\delta$.

*5. Line 142: 'hearth' should be 'heart'.*

We thank the Referee for spotting the typo, we corrected the text.

L146: Replace *"hearth"* with *"heart"*.

6. *Line 180: 'scientists is exposed' should be 'scientists are exposed'.*

We thank the Referee for spotting the typo, we corrected the text.

L183: Replace *"is"* with *"are"*.

7. *Line 237: 'it builds' should be 'does it build'.*

Thank you, we adapted the text according to the Referee's suggestion.

L240-241: Rephrase *"Not only it builds upon Sympl, but it also extends it with grid-aware and stencil-oriented functionalities."* as *"Not only **does it build** upon Sympl, but **the package also provides** grid-aware and stencil-oriented functionalities."*.

8. *Line 237: It's not clear to me what 'grid-aware' means in this context. Could you be specific?*

As we mention in the following sentences, this includes that all components are instantiated over a ComputationalGrid (cf. Listing 2) which collects information about the underlying (Cartesian) grid, e.g. grid spacings, index spaces.

No changes have been made to the text.

9. *Line 296: I don't think NPROMA has been defined.*

We thank the Referee for touching upon this aspect, which has been raised in other reviews as well. In the revised manuscript, we shortly describe what the NPROMA represents and refer the reader to available literature resources (e.g. Bauer et al., 2020; doi.org/10.21957/gdit22ulm, and Müller et al., 2019; doi.org/10.5194/gmd-12-4425-2019) for further details on the NPROMA slicing technique.

L299: Add footnote 13: *"NPROMA blocking is a cache optimization technique adopted in all Fortran codes considered in this paper. Given a two-dimensional array shaped (K∗M, N), this is re-arranged as a three-dimensional array shaped (K, M, N). Commonly, the leading dimension of the three-dimensional array is called "NPROMA", with K being the "NPROMA blocking factor". Here, we indicate K simply as*

*"NPROMA" for the sake of brevity. For further discussion of the NPROMA blocking, we refer the reader to Müller et al. (2019) and Bauer et al. (2020).".*

**RC3 (posted by Anonymous Referee #3 on 24.07.2024)**

*This is an excellent paper expanding the use of domain specific languages (DSLs), and GT4Py specifically, for performance and productivity in numerical weather prediction. To my knowledge this is the first published work on a tangent-linear or adjoint model in GT4Py, and the results are very encouraging. The authors describe their methodology and development process well, which will aid others looking to reproduce this work and apply it to their own models. That said I do have some small questions and comments I would like to raise before publication:*

We would like to express our sincere gratitude to the Referee for reviewing our work so carefully. We are very pleased to read the Referee's general appreciation of this study, and we thank them for the many constructive and valuable comments that we address point-by-point in the following.

**Primary points/questions:**

1. *I don't think it is necessary to define the tangent linear or adjoint operators explicitly, and I'm also not certain that you need to explicitly define the Taylor test either.*

   To the best of our knowledge, there does not exist any good reference online for the Taylor test and the symmetry test, so we would like to keep Algorithms 1 and 2. However, we agree with the Referee that the details of the tests are not essential for the main purposes of the paper, therefore we decided to move both Algorithms to the new Appendix A titled "Algorithmic description of the Taylor test and symmetry test for CLOUDSC2".

   - Move Algorithm 1 and 2 to the new Appendix A titled *"Algorithmic description of the Taylor test and symmetry test for CLOUDSC2"*.
   - L140-141: Add *"(Appendix A)"*.
   - L146: Add *"in Appendix A"*.
   - L415-417: Add *"In Section 2.2, we briefly described the aim and functioning of the Taylor test and the symmetry test for CLOUDSC2. Here, we detail the logical steps performed by the two tests with the help of pseudo-codes encapsulated in Algorithms A1 and A2."*.

2. *Line 247: I would like to see more description of the infrastructure code around the stencil. What does compile_stencil look like? Presumably the parent DiagnosticComponent class specifies the __call__ method, which wraps array_call, but that would be nice to see explicitly instead of assuming from what is in the paper.*

We are glad to read the Referee's interest in our infrastructure code. However, it is beyond the scope of the paper to describe the infrastructure code in detail, and we think that this pertains more to a technical documentation, rather than a scientific paper. We refer the Reviewer and any interested reader to the source code of ifs-physics-common (https://github.com/stubbiali/ifs-physics-common) and a talk given by the lead author of the paper at the recent PASC24 conference (https://event.pasc24-conference.org/slots/msa212).

No changes have been made to the text.

3. *Line 250: Similarly, the stencil collection decorator is ifs-specific, and I would appreciate more detail about what it does and how.*

Please refer to our reply to the previous point.

No changes have been made to the text.

4. *Line 265: Why use a GT4Py backend for CPU but a DaCe backend for gpu?*

As mentioned at L297-298 and now also at L299-301, for each programming paradigm (either in Fortran, C or Python) we only show performance numbers for the fastest variant. Since the GridTools CPU backend is found to be faster than the DaCe CPU backend, we only take into account the former. On the other hand, the GridTools GPU and DaCe GPU backends offer very similar performance on MeluXina and LUMI. However, we could not compile the CLOUDSC stencil using the GridTools GPU backend on Piz Daint, presumably because of a bug in CUDA11. We therefore decided to show results for the DaCe GPU backend only, as they were available on all three machines.

L299-301: Add *"Similarly, for all Python implementations we consider only the most performant backends of GT4Py: the GridTools C++ CPU backend with k-first memory layout, and the DaCe GPU backend."*.

5. *Line 345: Is the goal of the GT4Py or ECMWF teams to achieve the same performance as native Fortran and CUDA models, or is it to attain most of their performance alongside the benefits of portability and productivity?*

We aim for productivity and portability while achieving competitive performance on both GPU and CPU. As we mention in the revised paper, since the DSL can accommodate any specific low-level optimization, attaining the same performance as native Fortran and CUDA models is feasible and will be the target of our future efforts.

L400-402: Add the sentence *"Indeed, since the DSL can accommodate any specific low-level optimization, attaining the same performance as native, expert-written Fortran and CUDA/HIP models is feasible and will be the target of our future efforts."*.

6. *Figures 3-5: I'm not convinced by the layout of these figures. Because there are fewer implementations of CLOUDSC2 (and none in 32-bit aside from GT4Py) it may be more natural to report these performance results in a table, or to remove the space for the missing data, especially panels e and f which look disconcertingly sparse. On the other hand this is a very striking way to draw attention to the fact that GT4Py gives you 64- and 32-bit versions of the model in one go, but if you want to emphasize that I would like to see it more explicitly highlighted in the text.*

We are glad to read that the Referee finds the layout of Figs. 3-5 a striking way to emphasize the enhanced portability and flexibility of GT4Py codes. This is indeed our goal, therefore we prefer to keep the figures as they are. Moreover, we believe that these aspects are already sufficiently highlighted in the text.

No changes have been made to the text.

**Minor:**

1. *In your introduction is it worthwhile to discuss efforts to use tools like Numba or Cython to accelerate numerical models written in Python across various fields of science, such as Augier et al. (doi:10.1038/s41550-021-01342-y) or others?*

We thank the Referee for the meaningful suggestion. However, we note that it is beyond the scope of the paper to discuss how to accelerate Python codes in general. We believe that the Introduction is already very comprehensive.

No changes have been made to the text.

2. *Line 18: the authors describe Fortran's "functional programming style" which is slightly imprecise; while Fortran uses functions and subroutines, functional programming refers to a style of programming using only pure functions, so no values are updated in-place, which is not how Fortran operates.*

We thank the Referee for this clarification, which we agree with. In the revised manuscript, we define Fortran as a "procedural programming" language.

L18: Replace *"functional"* with *"procedural"*.

3. *Line 177: It would be useful to acknowledge contributions from groups beyond the Allen Institute, since they have ceased their work on GT4Py.*

Only major partners of ETH Zurich are mentioned; the only exception is the Allen Institute for Artificial Intelligence (AI2), for their significant contributions to GT4Py.Cartesian.

No changes have been made to the text.

4. *Line 190: "GTScript abstracts spatial for-loops away" would be more accurate than stating it abstracts for-loops entirely.*

Thank you, we adapted the text according to the Referee's suggestion.

L193: Rephrase *"GTScript abstracts for-loops away"* as *"GTScript abstracts **spatial** for-loops away"*.

5. *Line 237: "Not only it builds upon Sympl, but it also extends it" should be "Not only does it build upon Sympl, but also extends it".*

Thank you, we rephrased the sentence following the Referee's suggestion.

L240-241: Rephrase *"Not only it builds upon Sympl, but it also extends it with grid-aware and stencil-oriented functionalities."* as *"Not only **does it build** upon Sympl, but **the package also provides** grid-aware and stencil-oriented functionalities."*.

6. *Figure 6: Because the relevant information is contained within the top ~10% of the plot it may be useful to change the y-axis to instead range from 0.8 to 1.0.*

Although we understand the point made by the Referee, we find the plots in Fig. 6 to be more informative if the full bars are shown, so that one can better appreciate the fraction of the total runtime spent on the Python side.

No changes have been made to the text.

7. *Listing 1: Should foealfcu be "foealpha"?*

We thank the Referee for spotting that the "foealfcu" function was defined in Listing 1 instead of "foealfa". In the revised version, we include the definition of "foealfa" in Listing 1 and drop "foealfcu" (not used in the "saturation" stencil).

Listing 1, L2-4: Replace the "foealfcu" function with "foealfa".

**Other comments:**

1. *Line 323: The fact that the GT4Py implementations of the tangent-linear and adjoint formulations of CLOUDSC2 are the first to enable GPU execution at any precision is very cool and could be emphasized more heavily throughout the paper, in my opinion.*

We are grateful to the Referee for acknowledging and appreciating our work. However, we would like to point out that the primary purpose of the paper is to show the effectiveness of the DSL approach (and in particular GT4Py) for the NWP domain, with respect to productivity, portability, and GPU performance. We believe the robustness of the

GT4Py approach, including with respect to the TL/AD codes and single precision, becomes very clear from the current version of the manuscript.

No changes have been made to the text.

2. *Line 361: It might be worth mentioning that the Python overhead would still account for around 1% of CPU runtime even if the GT4Py CPU performance was on par with Fortran.*

We thank the Referee for noting this interesting fact, which is now highlighted in the text.

L371: Add *"; the latter corresponds to 0.7% relative to the Fortran execution time".*

**Additional changes**

- L15: Replace *"competitive performance"* with *"competitive **GPU** performance"*.
- Listing 1: Except for field offsets, all numerical values are now expressed as floats, instead of integers.
- L429-430: Add the statement *"We would like to thank three anonymous referees for carefully reviewing the manuscript and providing many constructive comments."*.
- L432-434: Add the statement *"CK acknowledges support from the ESiWACE3 project funded by the European High Performance Computing Joint Undertaking (EuroHPC JU) and the European Union (EU) under grant agreement No 101093054."*.
- L434: Add *"additionally"*.